# Association between the Physical Activity Behavioral Profile and Sedentary Time with Subjective Well-Being and Mental Health in Chilean University Students during the COVID-19 Pandemic

**DOI:** 10.3390/ijerph19042107

**Published:** 2022-02-13

**Authors:** Daniel Reyes-Molina, Jesús Alonso-Cabrera, Gabriela Nazar, Maria Antonia Parra-Rizo, Rafael Zapata-Lamana, Cristian Sanhueza-Campos, Igor Cigarroa

**Affiliations:** 1Facultad de Ciencias Sociales, Universidad de Concepción, Concepción 4030000, Chile; danielreyes@udec.cl; 2Departamento de Matemáticas y Estadística, Universidad del Norte, Barranquilla 081008, Colombia; jcabrera@uninorte.edu.co; 3Departamento de Psicología y Centro de Vida Saludable, Universidad de Concepción, Concepción 4030000, Chile; gnazar@udec.cl; 4Departamento de Psicología de la Salud, Universidad Miguel Hernández, 03202 Alicante, Spain; maria.parrar@umh.es; 5Facultad de Psicología, Universidad Internacional de Valencia, 46002 Valencia, Spain; 6Escuela de Educación, Universidad de Concepción, Los Ángeles 4440000, Chile; rafaelzapata@udec.cl (R.Z.-L.); crsanhuezac@udec.cl (C.S.-C.); 7Escuela de Kinesiología, Facultad de Salud, Universidad Santo Tomás, Los Ángeles 4440000, Chile

**Keywords:** COVID-19, physical activity, subjective well-being, mental health, university students

## Abstract

Objective: To analyze the association between the behavioral profile of physical activity and sedentary time with subjective well-being and mental health in university students during the COVID-19 pandemic in Chile. Methods: Cross-sectional study in a voluntary sample of 469 university students (22.4 ± 0.19 years; 66% women). According to students’ self-reports of physical activity and sedentary time, four behavioral profiles were created to investigate their association with subjective well-being and mental health using one-factor ANOVA that was adjusted to a multifactorial model. Results: The physically inactive and sedentary behavior profile presents the lowest levels of subjective well-being (*p* < 0.001), positive affective experiences (*p* < 0.001) and general mental health (*p* = 0.001). When adjusting for confounding variables, it was observed that the physically active and non-sedentary profile was associated with better general mental health (*p* < 0.01) in contrast to those who are physically active and sedentary. Conclusions: Chilean university students with a physically inactive and sedentary profile during the pandemic presented worse well-being and mental health, with a sedentary lifestyle being one of the variables that most affects the mental health of these students. Therefore, measures should be implemented to encourage this population to maintain adequate levels of physical activity and reduce sedentary times.

## 1. Introduction

Advances in urbanization and modernization in recent decades have generated important modifications in populations’ lifestyles, which are characterized by high levels of physical inactivity and sedentary behaviors [1,2,3,4,5]. In this regard, the coronavirus pandemic (COVID-19), recognized as one of the greatest health crises in the world, with unprecedented social and economic consequences, has had a negative impact on this panorama [6,7,8].

In Chile, with the arrival of the first cases of COVID-19 in March 2020, as in most countries worldwide, measures were implemented to mitigate the spread of the virus, which modified the activities of daily living in the entire population [9,10]. Specifically, the Chilean government implemented public health measures, such as restrictions on movement between cities and access to public places (fitness centers, cinemas, among others), social distancing, and dynamic periods of quarantine (i.e., while some cities could be in total or weekend quarantine, others left that status, depending on the local epidemiological situation) [11]. These measures led to the total closure of educational centers, including higher education institutions, forcing the university population to adapt to a mainly virtual learning environment [11,12], impacting on this group’s routines and lifestyles [13,14,15,16,17].

Consequently, during the COVID-19 pandemic, international studies reported a reduction in the practice of physical activity [18], and an increase in sedentary behavior in the university population [19], in addition to psychological health problems, such as, feelings of loneliness [20], perception of stress [21,22], anxiety [23] and depression symptoms [21].

The effects of the pandemic described above are of special concern, since prior to the pandemic, it has been described that the university population worldwide shows a high prevalence of physical inactivity [24], even much higher than the standardized global prevalence by age [2]. Similarly, a recent systematic review and meta-analysis study indicates that sedentary time in this population has increased in the last ten years, presenting an average of nine hours per day of sedentary behaviors [25].

According to the World Health Organization (WHO), a person who meets the recommendation of 150–300 min of moderate intensity of physical activity or 75–150 min of vigorous intensity per week is recognized as physically active [26]. On the other hand, those who spend more than four hours a day in activities, such as driving, sitting, lying down, or exposed to a screen time, can be classified as sedentary [27,28]. In this regard, being physically active and sedentary are not mutually exclusive opposites; on the contrary, as a consequence of current lifestyles, it is possible to comply with the recommendations of minimal physical activity per week, and also present sedentary times greater than four hours per day, which is known as the “physically active but sedentary” paradox [29].

A sedentary lifestyle and physical inactivity are associated with various physical and mental illnesses, such as depression and anxiety [30,31,32], both of which have shown increasing trends in university students in recent decades [33,34,35]. In this regard, recent studies have sought to understand how the combination of physical activity and sedentary behavior leads to certain health problems and, in particular, of mental health. Physically active college students with low sitting times have been reported to have a low prevalence of stress and depressive symptoms [36,37,38], lower risk of mental health problems and of better sleep quality [39]. From a positive psychological perspective, there is evidence supporting the association between physical activity and well-being, life satisfaction, and happiness in university students [40,41,42,43,44].

Currently, multinational studies in the COVID-19 context, carried out in different latitudes worldwide including Latin America and Chile, report a negative impact of the pandemic on levels of physical activity and sedentary time, being related to a worse mental health in the adult population [45,46,47,48,49,50,51]. These findings also apply to the university population. Studies, such as that of Coakley et al., which explored the association between physical activity and sedentary behavior with symptoms of depression and anxiety during COVID-19 restrictions, reported that students with low levels of physical activity and high sedentary times had greater symptoms of depression and anxiety than those who performed more physical activity and had less sitting time [52]. Other recent studies have suggested a positive association between university students who maintained or adopted a physically active behavior during the pandemic and subjective well-being [53,54].

However, the previous evidence that confirms the association between physical activity, mental health, and well-being is contrasted with the few studies that analyze this relationship using a combined profile of physical activity and sedentary time, in the context of a pandemic. Based on the above, this study set out to analyze the relationship between the behavioral profile of physical activity and sedentary time with subjective well-being and mental health in university students during the COVID-19 pandemic in Chile. We hypothesize that physically inactive and sedentary students have worse levels of subjective well-being and mental health than physically active and non-sedentary students (Hypothesis 1). Likewise, physically active and sedentary students present lower subjective well-being and mental health than their physically active and non-sedentary peers (Hypothesis 2).

## 2. Materials and Methods

### 2.1. Design and Participants

This is a correlational-causal and cross-sectional study in students from 24 higher education institutions of 13 regions of Chile, who were accessed by convenience through the snowball method by means of institutional emails and social networks (Facebook, Instagram, and WhatsApp). The participants had a mean age of 22.4 ± 0.19 years, 66% of which were women. A total of 61.5% of the students were in only weekend quarantined districts, and most reported not having contracted COVID-19 (88%), never using the sports time band (69.9%) (i.e., the hours from 5 to 9 am for individual outdoor physical activity, for cities that were in total or weekend quarantine [11]), 82.5% living in houses, and 88.2% having access to green areas.

The inclusion criteria were: (1) being a student at a Chilean higher education institution, and (2) being a Chilean or a foreigner residing in Chile. The exclusion criteria were: (1) being a postgraduate student (graduate, master’s, or doctorate); (2) being older than 29 years [55]; and (3) having reported a health and/or physical condition that prevented the engagement in physical activity during the previous six months. A total of 469 participants were recruited, of which 382 volunteered to be part of the sample, after having applied the exclusion criteria.

### 2.2. Procedure and Instruments

The participants responded an online questionnaire, available on the Google Forms platform, between 19 May and 30 June 2021, while several measures pertaining to the state of emergency, such as restriction of movement, dynamic quarantine periods and mass vaccination processes (2-dose vaccination schedule 21 days apart) [11], were still in place in Chile.

All participants agreed to be part of the study by signing an informed consent before answering the questionnaire. The procedures and methods used in the present study complied with the ethical guidelines defined by the Declaration of Helsinki [56], and were duly approved by the Ethics, Bioethics and Biosecurity Committee of the University of Concepción, Chile (CEBB 913-2021).

*Physical activity level and sedentary time:* The International Physical Activity Questionnaire (IPAQ) was used in its short version of seven items, which consists of a recall measure of seven days [57]. For the level of physical activity, participants must report the frequency (days per week) and duration (hours and minutes) of vigorous, moderate and light physical activity performed during the previous week [58]. The level of physical activity is classified into three levels (low, moderate, and high), based on the total metabolic equivalents (METs) per week, whose value corresponds to the sum of the METs of physical activity of light, moderate and vigorous intensity [58]. The METs for each intensity are obtained by multiplying the MET value (3.3 METs for light intensity, 4.0 METs for moderate intensity and 8.0 METs for vigorous intensity) by the total minutes per week of each type of intensity of physical activity [58,59]. For sedentary time, the participants were asked to report the hours and minutes spent sitting during a weekday (for example, in a class, at home, during free time, on the bus, watching television, etc.) [58]. This self-report questionnaire, validated and recommended to evaluate physical activity [57], has been previously applied in the university population in Chile [60].

Based on the surveyed students’ self-reports of physical activity level and sedentary time, a profile of physical activity behavior was created. Those who reported a low level of physical activity were considered physically inactive, while those who reported a moderate or high level of physical activity were considered physically active [58]. Additionally, they were classified as sedentary (sedentary time of >4 h per day) or non-sedentary (sedentary time of ≤4 h per day), according to the report of hours of sedentary behavior per day [27,28]. Based on these criteria, the participants were classified into four groups: (1) physically active and non-sedentary (PA-NS), (2) physically active and sedentary (PA-S), (3) physically inactive and non-sedentary (PI-NS), and (4) physically inactive and sedentary (PI-S).

Additionally, they were asked whether they used the sports time band, implemented by the Chilean government, to do outdoor physical activity during quarantine periods [11].

*Subjective well-being:* The ten-item subscale of subjective experienced well-being from the Pemberton Happiness Index (PHI) self-report questionnaire [61,62] was used. In this scale, participants must indicate their previous-day emotional experience, with a dichotomous answer (yes/no), for five items of positive affective experiences (e.g., “I did something that I really enjoyed doing”), and five items of negative affective experiences (e.g., “I felt belittled by someone”).

For experienced well-being, the total score of the PHI scale was considered, where the items are converted into a single score from zero (zero positive experiences and five negative experiences) to 10 (five positive experiences and no negative experiences). Scores of six or less indicate low experienced well-being, and scores of seven or more indicate high experienced well-being [62]. The dimensions of positive affective experiences and negative affective experiences were also analyzed individually. For the positive affective experiences score, only the 5 items referring to this dimension were considered, therefore, the score could range between 0 (zero positive experiences) and 5 (five positive experiences). Likewise, for the negative affective experience, only the 5 items referring to this dimension were considered; therefore, the score could range between 0 (zero negative experiences) to 5 (five negative experiences). The PHI has shown a validity higher than α *=* 0.82 and reliability of ω = 0.69 [61], having previously been used in Chile and presenting an internal consistency of α = 0.89 and reliability of ω = 0.90 [63].

*General mental health:* The 12-item version of the General Health Questionnaire (GHQ-12) [64] was used. This scale measures mental health through six items written positively (e.g., “Have you been able to concentrate well on what you are doing?”) and six written negatively (e.g., “Have you constantly felt overwhelmed and stressed?”) by means of a four-level Likert scale response format, going from 0 (Never) to 3 (Always). The score range is 0 to 36, with higher scores indicating poorer mental health [65], where scores equal to or less than 16 can be considered normal and scores greater than 16 can be considered high [66]. Scores above the cutoff point of 12 could be classified as poor mental health [67]. GHQ-12 has been previously validated in Chile, presenting a validity of α *=* 0.86 and reliability of ω = 0.90 [68,69].

*Mental health symptoms*: Four items were selected from the Patient Health Questionnaire (PHQ-9) [70], of the instrument adapted to Spanish by Barrigón et al. [71]. The four items selected to measure mental health symptoms were: (1) sleeping problems (“problems for falling asleep, staying a sleep or sleeping too much”); (2) fatigue (“feeling tired or having little energy”); (3) changes in eating behavior (“poor appetite or eating too much”); and (4) concentration problems (“problems concentrating on something, like reading the newspaper or watching television”). This instrument considers the frequency of a personal situation in the previous week, with a 4-level Likert scale response format (1 = never to 4 = almost every day).

*Sociodemographic information*: An ad hoc questionnaire was developed, which collected information on sociodemographic variables (gender and age), public health measures and COVID-19 variables (quarantine status by city, status of the vaccination process, infected by COVID-19 and symptoms of COVID-19), physical activity support variables (use of the sports time band, type of housing, access to green areas, and family income in Chilean pesos), and educational variables (study programs, years of study, and hours of study per day).

### 2.3. Statistical Analysis

Upon data collection on the different variables of the study—physical activity level and sedentary time, subjective well-being, mental health, and sociodemographic information—a descriptive statistical analysis is presented first. The qualitative data were represented by frequency and percentage, while the quantitative data by the mean and standard deviation. Data distribution was established by means of normality and equality variance tests (Shapiro–Wilk and Levene). The difference in means between two different groups was tested with the independent samples Student t-test. To determine the effect of profile of physical activity behavioral and sedentary time on the participants’ characteristics and subjective well-being and general mental health were determined using one-way ANOVA. (*** *p*-value < 0.001; ** *p*-value < 0.01; * *p*-value < 0.05). Cohen’s d effect size (ES) was calculated and qualitatively assessed as trivial (0–0.19), small (0.20–0.49), medium (0.50–0.79), or large (0.80 and greater) (Table 1 and Table 2, Figure 1).

Additionally, an adjusted multifactorial model was used to analyze the significance estimates of the cofactors main effects detected in the initial analysis. Data were presented as mean and its 95% CI. All analyses were incrementally adjusted according to different confounding factors. Model 0 was unadjusted; model 1 was adjusted for gender, symptoms of COVID-19, sports time band, access to green areas and study program (Table 1).

For both the unifactorial model and the multifactorial model, the following contrasts were applied: Bonferroni, to determine differences between the means and the fixed main effects, due to the four treatments or levels of the factor (*p* value < 0.05).

The chi-square test (χ^2^) was used to establish the association between mental health symptoms and behavior profile (Table 3). All the clean data were statistically treated and submitted to the respective analysis using the SPSS Statistic 27 (2020) software. Significance at the level of *p* < 0.05 was used.

## 3. Results

Table 1 shows descriptive data on the characteristics of the participants according to sociodemographic variables, public health and COVID measures, support for physical activity, and educational characteristics. In general, the university students were mostly women (66.0%), and most of the students reported ages between 18–20 years (34.8%). The students were mainly in a quarantined city only at weekends (61.5%), about a third of the students were inoculated with a first dose (29.6%) or a second dose of vaccination (37.4%), and had not contracted COVID-19 disease (88.0%). The majority never used the sports time band (69.9%), lived in a house (82.5%), had access to green areas (88.2%) and had a family income in Chilean pesos of 296,000–607,000. Most of the participants were studying some university degree related to health (46.3%), were in the third (22.8%) year, or between fifth and seventh year of the program (22.8%), and most reported studying between 6 and 8 hours a day (29.8%).

Additionally, Table 1 shows the significant differences according to the descriptive data of the participants’ characteristics in variables of experienced well-being, positive and negative affective experience, and general mental health. Specifically, in the sociodemographic variables, it is observed that women have a significantly higher average general mental health than men, which is interpreted as women having worse general mental health compared to men (17.77 ± 5.85 v/s 15.75 ± 6.67; *t* (380) = 9.242, *p* = 0.003, r^2^ = 0.42, with a small effect size). In public health measures and COVID-19 variables, when symptoms due to COVID-19 infection occurred, the group with mild symptoms had a significantly lower mean of positive affective experiences than the asymptomatic group (3.38 ± 1.25 v/s 4.44 ± 0.73; *t* (44) = 5.942, *p* = 0.019, r^2^ = 0.66, with a medium effect size).

Regarding the variables to support physical activity, those who sometimes used the sports time band had a mean of positive affective experiences significantly higher than that of the group that never used the sports time band (3.82 ± 1.14 v/s 3.13 ± 1.31; *t* (380) = 23.757, *p* < 0.001, r^2^ = 0.773, with a medium effect size), as well as significantly lower average general mental health than the group that never used the sports time band, which is interpreted as those who used the sports time band presented better general mental health than those who did not use sports time band (15.90 ± 5.97 v/s 17.59 ± 6.25; *t* (380) *=* 6.019, *p*
*=* 0.015, r^2^
*=* 0.295, with a small effect size). Similarly, those with access to green areas had significantly higher average experienced well-being than those without access to green areas (5.96 ± 1.67 v/s 5.00 ± 1.70; *t* (380) = 12.983, *p* < 0.001, r^2^ = 0.554, with a medium effect size), a significantly higher mean positive affective experience than those who did not have access to green areas (3.41 ± 1.25 v/s 2.82 ± 1.57; *t* (380) = 8.177, *p* = 0.004, r^2^ = 0.386, with a small effect size), and significantly lower average general mental health than those without access to green areas, which is interpreted to mean that those with access to green areas have better general mental health compared to those without access to green areas (16.68 ± 6.08 v/s 20.11 ± 6.36; *t* (380) *=* 12,500, *p* < 0.001, r^2^
*=* 0.539, with a medium effect size) (Table 1).

Regarding educational variables, it was found that those who studied a program in social sciences or humanities had a significantly lower average of positive affective experiences than those who were pursuing degrees in the areas of education, engineering/management, or health (2.94 ± 1.24 v/s 3.84 ± 1.16 v/s 3.35 ± 1.37 v/s 3.34 ± 1.29; *F* (3,378) *=* 5.679, *p*
*=* 0.001, η^2^
*=* 0.043, with a trivial effect size) (Table 1).

In Figure 1, it can be observed that university students belonging to the group with the PI-S behavioral profile, presented significantly lower means of experienced well-being (*p* < 0.001) and significantly lower means of positive affective experiences (*p* < 0.001) than the other three groups of behavioral profiles. Likewise, the group with the PI-S behavioral profile had a significantly higher mean general mental health score (*p* = 0.001) than the other three groups of behavioral profiles, which is interpreted as the PI-S group presenting worse general mental health than the other three groups.

Table 2 shows a non-adjusted multifactorial model (Model 0) and an adjusted one (Model 1). Both models analyze the significant differences of the main effects for the variables of experienced subjective well-being, positive and negative affective experience, and general mental health from the behavioral profiles of physical activity and sedentary time.

For the non-adjusted multifactorial model (Model 0), behavioral profiles PA-NS (*p* < 0.001), PA-S (*p* < 0.001), and PI-NS (*p* < 0.05) showed higher levels of experienced well-being and positive affect than university students who had a PI-S behavioral profile. In the case of negative affective experiences, no significant differences were reported between the behavioral profiles of physical activity and sedentary time. On the other hand, the university students with behavioral profiles PA-NS (*p* < 0.001), and PA-S (*p* < 0.01) showed better general mental health than the university students with the PI-S behavioral profile (Table 2).

For the adjusted multifactorial model (Model 1), the significant differences and confidence intervals of the main effects were analyzed from the cofactors (gender, symptoms of COVID-19, sports time band, access to green areas and study program) detected in the initial analysis (Table 1). In this sense, university students with behavioral profiles PA-NS (*p* < 0.001), PA-S (*p* < 0.001), and PI-NS (*p* < 0.05) showed higher levels of experienced well-being and positive affect than university students who had a PI-S behavioral profile. Additionally, university students with a PA-NS (*p* < 0.001) showed with better general mental health than the university students with a PI-S behavioral profile (Table 2).

Table 3 shows that the behavior profiles of physical activity and sedentary time were significantly related to mental health symptoms such as fatigue (χ^2^(9) = 32.922, *p* < 0.001, V de Cramer = 0.169), changes in eating behavior (χ^2^(9) = 40.451, *p* < 0.001, V de Cramer = 0.188), concentration problems (χ^2^(9) = 17.587, *p* = 0.040, V de Cramer = 0.124). PI-S was the profile that reported the highest prevalence (almost every day) of fatigue (43.3%), appetite disorders (45.4%) and concentration problems (35.1%). Additionally, it was observed that the non-sedentary behavior profiles (PA-NS and PI-NS) reported a higher prevalence in terms of the absence of mental health symptoms (never), than the sedentary profiles (PA-S and PI-S). Likewise, it was observed that the profiles with sedentary behavior (PA-S and PI-S) reported a higher prevalence of mental health symptoms (almost every day) than the non-sedentary profiles (PA-NS and PI-NS).

## 4. Discussion

This study aimed to analyze the relationship between the behavioral profile of physical activity and sedentary time with subjective well-being and mental health in Chilean university students during the COVID-19 pandemic. The main findings suggest that Chilean university students who presented a physically inactive and sedentary profile during the COVID-19 pandemic experienced worse well-being, positive affective experiences, and general mental health. When adjusting for confounding variables, students who presented a physically active and non-sedentary profile were associated with better general mental health than those who are physically active and sedentary. Additionally, in relation to mental health symptoms, the behavior profile was related to fatigue, changes in eating behavior, and concentration problems, presenting a higher prevalence in students with sedentary behavior profiles regardless of the behavior related to physical activity. On the contrary, those who had a physically active behavior profile presented a lower prevalence of mental health symptoms regardless of their sedentary time.

These findings support hypothesis 1 raised in this study: physically inactive and sedentary college students have worse levels of subjective well-being and mental health compared to their physically active and non-sedentary peers. Similarly, these results are consistent with studies on the behavior of physical activity and sedentary time and its association with well-being and mental health in university students during the COVID-19 pandemic [40,45,52,72]. In this area, a study by Pengpid et al., in 12,492 university students from 24 countries, reported that physically active and non-sedentary students presented better life satisfaction, happiness and perceived health status during the COVID-19 pandemic [40]. Likewise, a study of 255 university students in the United Kingdom during the COVID-19 pandemic showed how well-being and physical activity decreased, while stress and a sedentary lifestyle increased [72].

Regarding hypothesis 2 raised in this study, the results do not provide sufficient evidence to support that physically active and sedentary student presented lower subjective well-being and mental health than their physically active and non-sedentary peers. However, when analyzing mental health symptoms, differences were observed between these behavioral profiles. Thus, physically active and non-sedentary students had a lower prevalence of mental health symptoms, such as fatigue, changes in eating behavior, and concentration problems. The foregoing, in contrast to what was presented by active but sedentary students, who presented a higher prevalence of these mental health symptoms. In this sense, similar results were observed by Rees-Punia et al. in which participants who increased their sedentary lifestyle, became inactive, or decreased their moderate to vigorous activity were more likely to experience depression related to psychological distress during the COVID-19 pandemic [73].

In addition, a particular finding could be observed regarding the mental health outcomes of both sedentary and non-sedentary physically active students, who reported significantly better general mental health than inactive and sedentary students. Specifically, a protective effect of physical activity on the mental health of physically active but sedentary students was observed. However, this protective effect disappears when considering the confounding variables, gender, symptoms of COVID-19, sports time band, access to green areas, and study program, whereas only physically active and non-sedentary students continue to maintain significantly better general mental health than inactive and sedentary students. In this sense, the study by Haider et al. found that the presence of high levels of physical activity was associated with greater well-being and fewer symptoms of depression and anxiety in times of the COVID-19 pandemic [74]. Furthermore, Fornili et al. reported that, during confinement, severe levels of anxiety or depression were found in 240,000 students with a high probability of being severely anxious or depressed for those who stopped practicing their usual practice of physical activity [75].

However, despite the aforementioned studies on university students investigated in the COVID-19 pandemic on physical activity and mental health, our study, carried out in Chilean university students with a comparative design of four groups (behavioral profiles), provides findings that support the effect of sedentary lifestyle as one of the variables that most affect mental health. Therefore, the scientific strength of the present publication lies in its design, by contributing new knowledge to the literature as the first study that considers the comparison between physical activity and sedentarism in young university students in the times of COVID-19. Thus, for the first time, it is possible to report that, compared to the practice of physical activity, sedentarism or physical inactivity causes the greatest impact on well-being and mental health care of young university students in times of COVID-19 pandemic.

Derived from this, one of the theoretical implications for the scientific community is the need to broaden the perspective of this area of study, prioritizing the research of sedentary time and its repercussions on mental health as an adverse effect. A better understanding of the effects of a sedentary lifestyle could shed more light on its repercussions for the mental health of young people, prevent this behavior, and foster others that promote mental health. On the other hand, one of the practical implications of the present study for society lies in providing updated evidence on the detrimental effects of physical inactivity and sedentary behavior on the emotional state of university students. These findings can serve as a guide for university authorities to promote programs to detect risk factors in the mental health of students, as well as educational programs aimed at not minimizing the adverse effects of sedentary lifestyle and its consequences on university students’ various emotional states. The information that can be provided to society on the effects of sedentary lifestyles on mental health would be the fundamental axis of health care education of young people.

### Strengths and Limitations

This study is the first to analyze the behavioral profiles of physical activity and sedentary time with subjective well-being and mental health in university students from different universities and regions of Chile during the COVID-19 pandemic. In this sense, the present study is a contribution to the knowledge about how the behavioral profile of physical activity and sedentary lifestyle can differentially modulate well-being and mental health. In addition, this study allowed generating a pattern of subjective well-being and mental health according to the different behaviors of physical activity and sedentary lifestyle of university students, which could help understand which groups of students have the worst indicators of well-being and mental health. Finally, this study contributes to understanding the role of physical activity and sedentary lifestyle in well-being and mental health, providing theoretical and empirical support to generate programs to promote physical activity and reduce sedentary lifestyle in the university population.

Nevertheless, the present study is not without its limitations. First, due to the public health measures derived from the pandemic, the variables studied were self-reported through the internet, specifically self-reported physical activity is subject to overestimation compared to objective measures, such as accelerometry [76]. Second, the cross-sectional design does not allow determining causal inferences about relationships between variables. Third, due to the convenience sampling method, selection bias is also a possibility. Fourth, even though this study considers different sociodemographic variables, well-being and mental health are constructs that can be modulated by multiple factors that were not measured in this study, such as family and academic relationships. Finally, this study does not distinguish between planned and unplanned (incidental) physical activity, given the motivational and behavioral differences between these types of physical activity, it is plausible that they are associated with differences in well-being and mental health [46].

One of the future lines of research, considering the design of this study, is to delve into the effects on well-being and mental health that the behavioral profile of physical activity and sedentary lifestyle could have in different populations with or without pathologies of physical and psychological health, as well as in different age groups. Similarly, it is convenient to continue studying the relationship between these variables, but using objective measurement instruments that complement the information provided by the self-report questionnaires [43,76]. For example, measurements of facial and non-verbal expression in the case of well-being [43], and for physical activity the use of accelerometers, which would allow a more precise recording of planned and unplanned physical activity [76]. Another aspect is the study of how much the different ways of coping with confinement may have affected the behavioral profile of active activity and sedentary lifestyle of university students.

## 5. Conclusions

The subjective well-being and mental health of Chilean university students was affected differently according to the profile of physical activity and sedentary time that the students presented during the COVID-19 pandemic. In this line, this study is the first to make a comparison of this type of behavior profiles in university students in the context of the COVID-19 pandemic. Out of the four behavioral profiles, university students who presented a physically inactive and sedentary one had worse experienced well-being, positive affective experiences, and general mental health. Additionally, students who presented a physically active and non-sedentary profile were associated with better general mental health than those who were physically active and sedentary. In this area, increasing physical activity and reducing sedentary lifestyle can promote well-being and mental health in the university student population during the COVID-19 pandemic. In particular, within the behavioral profiles, a sedentary lifestyle is one of the variables that most affect the mental health of Chilean university students. While physical activity is beneficial for mental health, as reported in the literature and in the present study. Therefore, awareness on the harmful effects that a sedentary lifestyle causes on the mental health of young university students is essential. Thus, in times of a health crisis, government and university leaders must implement measures to encourage this population to maintain adequate levels of physical activity, as well as reduce sedentary times.

## Figures and Tables

**Figure 1 ijerph-19-02107-f001:**
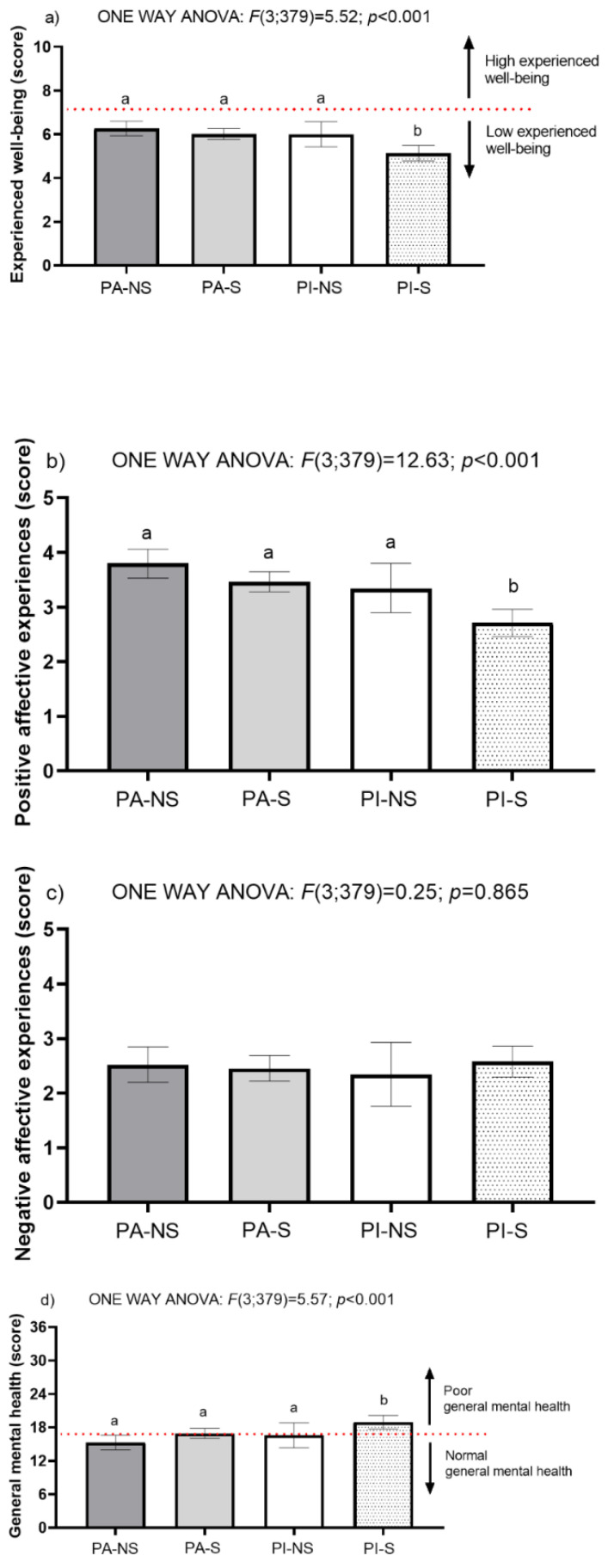
Subjective well-being and general mental health variables according to the physical activity behavioral profile and sedentary time. (**a**) Experienced well-being according to physical activity behavioral and sedentary time, (**b**) Positive affective experience according to physical activity behavioral and sedentary time, (**c**) Negative affective experience according to physical activity behavioral and sedentary time, and (**d**) General mental health according to physical activity behavioral and sedentary time. Note: PA-NS = physically active and non-sedentary, *n* = 84; PA-S = physically active and sedentary, *n* = 172; PI-NS = physically inactive and non-sedentary, *n* = 29; PI-S = physically inactive and sedentary, *n* = 97. Scores of six or less indicate low experienced well-being, and scores of seven or more indicate high experienced well-being. Regarding the general mental health variable on the GHQ-12 scale, the higher the score, the worse general mental health, where scores equal to or less than sixteen can be considered normal and scores greater than sixteen can be considered high. Variables are presented as mean and their respective 95% confidence interval.

**Table 1 ijerph-19-02107-t001:** Descriptive data of the participants’ characteristics.

	TOTAL N (%)	Experienced Well-Being M (SD)	Positive Affective Experience M (SD)	Negative Affective Experience M (SD)	General Mental Health M (SD)
**Sociodemographic variables**					
Gender					
*Women* *Men*	252 (66.0%) 130 (34.0%)	5.81 (1.72) 5.92 (1.67)	3.27 (1.29) 3.47 (1.31)	2.46 (1.46) 2.55 (1.59)	17.77 (5.85) 15.75 (6.67) **
Age					
*18–20 years* *21–22 years* *23–29 years*	133 (34.8%) 117 (30.6%) 132 (34.6%)	6.06 (1.79) 5.59 (1.53) 5.86 (1.74)	3.44 (1.27) 3.34 (1.29) 3.23 (1.33)	2.38 (1.61) 2.75 (1.45) 2.37 (1.41)	16.59 (6.08) 17.84 (6.15) 16.91 (6.35)
**Public health measures and COVID variables**					
Quarantine status per city					
*Total quarantine* *Weekend quarantine* *No quarantine*	103 (27.0%) 235 (61.5%) 44 (11.5%)	5.92 (1.60) 5.87 (1.76) 5.55 (1.64)	3.59 (1.18) 3.23 (1.34) 3.30 (1.25)	2.67 (1.55) 2.37 (1.48) 2.75 (1.48)	17.20 (6.15) 16.89 (6.38) 17.82 (5.44)
Vaccination process status					
*Non-vaccinated* *One-dose vaccinated* *Two-dose vaccinated*	126 (33.0%) 113 (29.6%) 143 (37.4%)	5.64 (1.71) 5.80 (1.44) 6.06 (1.86)	3.33 (1.40) 3.37 (1.27) 3.32 (1.24)	2.68 (1.57) 2.58 (1.51) 2.26 (1.42)	17.56 (6.02) 17.54 (6.28) 16.31 (6.27)
Infected by COVID-19					
*Non-infected* *infected*	336 (88.0%) 46 (12.0%)	5.86 (1.70) 5.76 (1.78)	3.30 (1.30) 3.59 (1.24)	2.45 (1.50) 2.83 (1.54)	17.23 (6.31) 16.02 (5.30)
Symptoms of COVID-19					
*Asymptomatic* *Mild symptoms*	9 (19.5%) 37 (80.5%)	6.33 (1.32) 5.68 (1.19)	4.44 (0.73) 3.38 (1.25) *	3.11 (1.96) 2.70 (1.50)	13.56 (4.40) 16.43 (5.27)
**Physical activity support variables**					
Use of the sport time band					
*Never used* *Sometimes used*	267 (69.9%) 115 (30.1%)	5.76 (1.81) 6.05 (1.41)	3.13 (1.31) 3.82 (1.14) ***	2.37 (1.48) 2.77 (1.54)	17.59 (6.25) 15.90 (5.97) *
Type of home					
*House* *Apartment* *Countryside house*	315 (82.5%) 33 (8.6%) 34 (8.9%)	5.87 (1.71) 5.94 (1.73) 5.50 (1.62)	3.34 (1.31) 3.12 (1.14) 3.53 (1.37)	2.47 (1.52) 2.18 (1.46) 3.03 (1.24)	17.08 (6.26) 17.33 (6.72) 16.88 (5.27)
Green areas access					
*No green areas access* *green areas access*	45 (11.8%) 337 (88.2%)	5.00 (1.70) 5.96 (1.67) ***	2.82 (1.57) 3.41 (1.25) ***	2.82 (1.46) 2.45 (1.50)	20.11 (6.36) 16.68 (6.08) ***
Family income in Chilean pesos (CLP)					
*<296,000* *296,000–607,000* *608,000–1,572,999* *>1,573,000*	56 (14.7%) 145 (38.0%) 124 (32.4%) 57 (14.9%)	5.59 (1.74) 5.83 (1.74) 5.81 (1.72) 6.19 (1.53)	3.09 (1.35) 3.41 (1.32) 3.34 (1.24) 3.40 (1.32)	2.50 (1.39) 2.57 (1.49) 2.52 (1.50) 2.21(1.68)	18.16 (5.96) 17.32 (5.88) 16.90 (6.65) 15.81 (6.17)
**Educational variables**					
Study programs					
*Education*	61 (16.0%)	6.11 (1.67)	3.84 (1.16)	2.72 (1.63)	17.00 (6.67)
*Engineering/Management*	66 (17.3%)	5.53 (1.68)	3.35 (1.37)	2.82 (1.56)	16.92 (6.16)
*Health area*	177 (46.3%)	5.98 (1.79)	3.34 (1.29)	2.36 (1.48)	17.08 (6.17)
*Social Sc./Humanities*	78 (20.4%)	5.59 (1.63)	2.94 (1.24) **	2.35 (1.35)	17.29 (6.06)
Study years					
*1* *2* *3* *4* *5 to 7*	57 (14.9%) 73 (19.1%) 87 (22.8%) 78 (20.4%) 87 (22.8%)	5.75 (1.79) 5.74 (1.60) 6.05 (1.78) 5.71 (1.59) 5.92 (1.76)	3.26 (1.39) 3.41 (1.22) 3.34 (1.27) 3.47 (1.21) 3.20 (1.40)	2.51 (1.71) 2.67 (1.47) 2.30 (1.55) 2.77 (1.43) 2.28 (1.37)	17.19 (5.37) 17.53 (6.43) 17.15 (6.33) 17.35 (5.93) 16.33 (6.69)
Study hours per day					
*1–3* *4–5* *6–8* *9–12*	111 (29.1%) 91 (23.8%) 114 (29.8%) 66 (17.3%)	5.68 (1.69) 5.99 (1.81) 5.98 (1.59) 5.70 (1.75)	3.18 (1.31) 3.45 (1.34) 3.35 (1.19) 3.42 (1.40)	2.50 (1.58) 2.46 (1.56) 2.37 (1.45) 2.73 (1.39)	17.41 (6.25) 17.15 (6.11) 16.72 (5.87) 17.08 (6.89)

Note. The n of each group was presented as frequency and percentage. The variables of experienced well-being, positive affective experience, negative affective experience, and general mental health as mean and standard deviation. Regarding the general mental health variable on the GHQ-12 scale, the higher the score, the worse general mental health. *** *p*-value < 0.001; ** *p*-value < 0.01; * *p*-value < 0.05.

**Table 2 ijerph-19-02107-t002:** Estimated effect of physical activity behavioral profile and sedentary time.

	Model 0 Non-Adjusted	Model 1 Adjusted
Variables	β_i_ [CI 95%]	β_i_ [CI 95%]
Experienced well-being		
PA-NS	1.14 [0.65; 1.62] ***	1.08 [4.48; 5.73] ***
PA-S	0.88 [0.46; 1.29] ***	0.86 [0.44; 1.28] ***
PI-NS	0.87 [0.18; 1.55] *	0.73 [0.05; 1.42] *
PI-S	Reference	Reference
Positive affective experience		
PA-NS	1.08 [0.72; 1.45] ***	0.84 [0.47; 1.20] ***
PA-S	0.75 [0.44; 1.06] ***	0.64 [0.33; 0.95] ***
PI-NS	0.63 [0.12; 1.15] *	0.52 [0.02; 1.02] *
PI-S	Reference	Reference
Negative affective experience		
PA-NS	−0.054 [−0.50; 0,39]	−0.23 [−0.70; 0.22]
PA-S	−0.124 [−0.50; 0.25]	−0.22 [−0.61; 0.16]
PI-NS	−0.232 [−0.86; 0.40]	−0.21 [−0.84; 0.41]
PI-S	Reference	Reference
General mental health		
PA-NS	−3.67 [−5.46; −1.88] ***	−2.84 [−4.7; −0.98] **
PA-S	−2.02 [−3.54; −0.50] **	−1.41 [−2.97; 0.15]
PI-NS	−2.36 [−4.9; 0.18]	−2.24 [−4.76; 0.28]
PI-S	Reference	Reference

Note. PA-NS *=* physically active and non-sedentary; PA-S *=* physically active and sedentary; PI-NS *=* physically inactive and non-sedentary; PI-S *=* physically inactive and sedentary. The variables are presented with the β_i_ coefficient and their respective 95% CI. Model 0 was not adjusted; model 1 was adjusted for gender, symptoms of COVID-19, sports time band, access to green areas, and study program. *** *p*-value < 0.001; ** *p*-value < 0.01; * *p*-value < 0.05.

**Table 3 ijerph-19-02107-t003:** Mental health symptoms according to the physical activity profile and sedentary time.

**Sleeping Problems**	**PA-NS** ** *n* ** **(%)**	**PA-S** ** *n* ** **(%)**	**PI-NS** ** *n* ** **(%)**	**PI-S** ** *n* ** **(%)**	***p*-Value**	**V de Cramer**
*Never*	21 (25.0)	35 (20.3)	6 (20.7)	9 (9.3)	-	0.118
*A couple of days*	35 (41.7)	67 (39.0)	10 (34.5)	32 (33.0)		
*Several days of the week*	12 (14.3)	23 (13,4)	5 (17.2)	23 (23.7)		
*Almost every day*	16 (19.0)	47 (27.3)	8 (27.6)	33 (34.0)		
**Fatigue**	**PA-NS** ** *n* ** **(%)**	**PA-S** ** *n* ** **(%)**	**PI-NS** ** *n* ** **(%)**	**PI-S** ** *n* ** **(%)**		
*Never*	11 (13.1)	15 (8.7)	3 (10.3)	3 (3.1)	***	0.169
*A couple of days*	41 (48.8)	65 (37.8)	13 (44.8)	20 (20.6)		
*Several days of the week*	20 (23.8)	48 (27.9)	6 (20.7)	32 (33.0)		
*Almost every day*	12 (14.3)	44 (25.6)	7 (24.1)	42 (43.3)		
**Changes in Eating Behavior**	**PA-NS** ** *n* ** **(%)**	**PA-S** ** *n* ** **(%)**	**PI-NS** ** *n* ** **(%)**	**PI-S** ** *n* ** **(%)**		
*Never*	23 (27.4)	33 (19.2)	11 (37.9)	9 (9.3)	***	0.188
*A couple of days*	38 (45.2)	52 (30.2)	9 (31.0)	22 (22.7)		
*Several days of the week*	12 (14.3)	34 (19.8)	5 (17.2)	22 (22.7)		
*Almost every day*	11 (13.1)	53 (30.8)	4 (13.8)	44 (45.4)		
**Concentration Problems**	**PA-NS** ** *n* ** **(%)**	**PA-S** ** *n* ** **(%)**	**PI-NS** ** *n* ** **(%)**	**PI-S** ** *n* ** **(%)**		
*Never*	16 (19.0)	26 (15,1)	3 (10.3)	8 (8.2)	**	0.124
*A couple of days*	36 (42.9)	54 (31.4)	14 (48.3)	33 (34.0)		
*Several days of the week*	20 (23.8)	45 (26.2)	8 (27.6)	22 (22.7)		
*Almost every day*	12 (14.3)	47 (27.3)	4 (13.8)	34 (35.1)		

Note. PA-NS = physically active and non-sedentary, *n* = 84; PA-S = physically active and sedentary, *n* = 172; PI-NS = physically inactive and non-sedentary, *n* = 29; PI-S = physically inactive and sedentary, *n* = 97. *** *p*-value < 0.001; ** *p*-value < 0.01; * *p*-value < 0.05.

## Data Availability

Data are available upon request.

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
