# Peer review of "Association between the Physical Activity Behavioral Profile and Sedentary Time with Subjective Well-Being and Mental Health in Chilean University Students during the COVID-19 Pandemic"

_ijerph, 2022, doi:10.3390/ijerph19042107_

Round 1

Reviewer 1 Report

  1. At least the affiliations of the authors must be in English.
  2. What is a “sports time band”? It is not widely known internationally. 
  3. Line 123, what are the dynamic quarantine periods? These are presented in Table 1 but there is no definition.
  4. Line 189 and Table 1, how were the positive and negative affective experiences defined and calculated?
  5. Methodology needs to be enriched with more information. For instance, Section 2.3 is too concise, so missing important information, e.g., Line 192, the adjusted multifactorial model needs proper description and reference, or why do we need to do the analysis resulting in Table 2 and 4. (Table 3 is missing).
  6. Section 3, vs or v/s are used interchangeably. Need correction.
  7. Before ‘public health measures” is mentioned in Table 1, it must be well described in the text. What kind of measures were undertaken in Chile in response to COVID-19 pandemic?  What are the COVID-19 variables considered in this study?
  8. The study was based on PHI, GHQ-12, PHQ-9 as described in Section 2.2. It is strongly needed to justify the applicability of these measure items to the context being studied, i.e. young university students in Chile. It is quite difficult to judge how these were well accepted and used generally, e.g., appeared as [60, 61]. The list of items may be provided in the appendix for interested readers. 
  9. The results in Table 1 and 2 were not well explained in the paragraph. It is suggested to interpret and explain them in an orderly manner. Short conclusion will be useful as how it was judged from the results in Table 1 and 2 respectively. 
  10. Presentation of the content in Figure 1 and Table 2 should be consistent. For example, well-being is presented as PHI in Figure 1a but as Experience well-being in Table 2. 
  11. Figure 1, what is F(3;591)? What is 591? 
  12. PANAS appears in Table 2 without proper definition. 
  13. Explanation of Table 2 is too coarse. 
  14. Section 4 (around lines 277 to 315) sounds more like a literature review than discussion. 
  15. In my opinion, line 320, it sounds too ambitious to claim the scientific strength of the study in the area (supposed to be in Latin America, or specifically Chile). It has been present in other studies but just in other geographical areas. 
  16. Chi-squared should be the Greek alphabet rather than X in the paragraph above Table 4.
  17. Line 339 to 341, 367 to 370, and 383 to 385 overly repeat the preceding sections. 
  18. Conclusions in Section 5 are superficial and not emphasizing the research contribution, therefore need a major revise.
  19. Generally speaking, the manuscript must be thoroughly revised to be more concise and scientifically rigorous. 

Author Response

Comment 1. At least the affiliations of the authors must be in English.

Response 1: As the reviewer correctly mentions, the study authors' affiliations are not in English. This is due to the requirements of each university since these universities are located in Spanish-speaking countries.

Comment 2. What is a “sports time band”? It is not widely known internationally. 

Response 2: We appreciate the reviewer's comment, which allowed us to better explain the concept of "sports time band". This measure that was implemented by the Government of Chile, in response to the difficulty in carrying out physical activity because of the initial isolation measures and movement restrictions applied during the first waves of the COVID-19 pandemic (from early 2020 to mid-2021). For this reason, in the first paragraph of section 2.1 Design and participants, the following definition was added that complements the concept that we indicated in the writing as "sports time band ". Now, after “sports time band”, it says:

(i.e., hours from 5 to 9 am for individual outdoor physical activity, for cities that were in total or weekend quarantine [11])”. Likewise, to improve the writing and understanding of the same paragraph, the following sentence was modified, where it said: “living in houses (82.5%), and having access to green areas (88.2%)”, now it says: “82.5% living in houses, and 88.2% having access to green areas.

Comment 3. Line 123, what are the dynamic quarantine periods? These are presented in Table 1 but there is no definition.

Response 3: Based on reviewer comment, we have added a definition to the concept "dynamic quarantine periods". Also, to respond to comment 8 made by the reviewer, this definition was added in paragraph 2 of section 1 Introduction of the manuscript, as follows:

“(i.e., while some cities could go into total or weekend quarantine, others left that status, depending on the local epidemiological situation), “

Comment 4. Line 189 and Table 1. How were the positive and negative affective experiences defined and calculated?

Response 4: We appreciate the reviewer's comment that allowed us to better define how positive and negative affective experiences were calculated using the Pemberton Happiness Index (PHI) self-report questionnaire. Therefore, in addition to specifying how the PHI score was calculated, we added a sentence that specifies how the positive and negative affective experiences score was obtained. Consequently, at the end of the third paragraph on page 8 belonging to section 2.2 procedure and instruments, referring to subjective well-being, the following is added:

“For experienced well-being, the total score of the PHI scale was considered, where the items are converted into a single score from zero (zero positive experiences and five negative experiences) to 10 (five positive experiences and no negative experiences). Scores of six or less indicate low experienced well-being, and scores of seven or more indicate high experienced well-being [62]. The dimensions of positive affective experiences and negative affective experiences were also analyzed individually. For the positive affective experiences score, only the 5 items referring to this dimension were considered, therefore, the score could range between 0 (zero positive experiences) and 5 (five positive experiences). Likewise, for the negative affective experience, only the 5 items referring to this dimension were considered, therefore, the score could range between 0 (zero negative experiences) to 5 (five negative experiences).”

Comment 5. Methodology needs to be enriched with more information. For instance, Section 2.3 is too concise, so missing important information, e.g., Line 192, the adjusted multifactorial model needs proper description and reference, or why do we need to do the analysis resulting in Table 2 and 4. (Table 3 is missing).     

Response 5: Based on comment 8, as well as other important comments made by the reviewer, we have been able to enrich the methodology section of our manuscript. In particular, section 2.3 Statistical analysis, had important modifications from this comment, with the purpose of strengthening and clarifying the statistical procedures that were carried out for the analysis of results. Consequently, now the paragraphs on page 5 of the manuscript, referring to section 2.3 of Statistical Analysis, read as follows:

“Upon data collection on the different variables of the study: physical activity level and sedentary time, subjective well-being, mental health, and sociodemographic information, a descriptive statistical analysis is presented first. The qualitative data were represented by frequency and percentage, while the quantitative data by the mean and standard deviation. Data distribution was established by means of normality and equality variance tests (Shapiro-Wilk and Levene). The difference in means between two different groups was tested with the independent samples T-Student test. To determine the effect of profile of physical activity behavioral and sedentary time on the participants’ characteristics and subjective well-bing and general mental health were determined using one-way ANOVA. (*** p-value <0.001; ** p-value <0.01; * p-value <0.05). Cohen’s d effect size (ES) was calculated and qualitatively assessed as trivial (0–0.19), small (0.20–0.49), medium (0.50–0.79), or large (0.80 and greater) (Table 1, Figure 1, and Table 2).

Also, an adjusted multifactorial model was used to analyze the significance estimates of the cofactors main effects detected in the initial analysis. Data were presented as mean and its 95% CI. All analyses were incrementally adjusted according to different con-founding factors. Model 0 was unadjusted; Model 1 was adjusted for gender, symptoms of COVID-19, sports time band, access to green areas and study program (Table 1).

For both the unifactorial model and the multifactorial model, the following contrasts were applied: Bonferroni, to determine differences between the means and the fixed main effects, due to the four treatments or levels of the factor (p value <0.05).

 The Chi-square test (χ2) was used to establish the association between mental health symptoms and behavior profile (Table 3). All the clean data were statistically treated and submitted to the respective analysis using the SPSS Statistic 27 (2020) software. Significance at the level of p < 0.05 was used.”

We have corrected the enumeration error regarding the tables in the results section of our manuscript. Table 4 of Mental health symptoms according to physical activity profile and sedentary time, became “Table 3

Comment 6. Section 3, vs or v/s are used interchangeably. Need correction.

Response 6: In accordance with this comment, the statement "v/s" was unified in the first two paragraphs of section 3 Results of this manuscript, where it was used. This has allowed us to improve the writing of the results section.

Comment 7. Before ‘public health measures” is mentioned in Table 1, it must be well described in the text. What kind of measures were undertaken in Chile in response to COVID-19 pandemic?  What are the COVID-19 variables considered in this study?

Response 7: We fully agree with the reviewer's comment on the public health measures implemented by the Government of Chile, which are mentioned in Table 1, and should be clarified in advance in the text. For this reason, in the introductory section of the manuscript, specifically in the second paragraph, where the public health measures implemented by the Government of Chile are mentioned, we have decided to deepen and specify in this regard. The second paragraph, page 2 of section 1. Introduction, now reads as follows:

“In Chile, with the arrival of the first cases of COVID-19 in March 2020, as in most countries worldwide, measures were implemented to mitigate the spread of the virus, which modified the activities of daily living in the entire population [9,10]. Specifically, the Chilean government implemented public health measures such as restrictions on movement between cities and access to public places (fitness centers, cinemas, restaurant, among others), social distancing, and dynamic periods of quarantine (i.e., while some cities could go into total or weekend quarantine, others left that status, depending on the local epidemiological situation) [11]. These measures led to the total closure of educational centers, including higher education institutions, forcing the university population to adapt to a mainly virtual learning environment [11,12], impacting on this group’s routines and lifestyles [13–17].”

With the purpose of providing clearer information on the Public Health measures and the COVID variables consulted in the online survey, and which were used in the study. As well as, to give consistency with what is mentioned in Table 1 of the manuscript, the information provided in the section "Sociodemographic information" was improved. Likewise, in section 2.2 Procedure and instruments, page 4-5 of the manuscript, the paragraph referring to "Sociodemographic information" was modified, thus remaining:

“Sociodemographic information: An ad hoc questionnaire was developed, which collected information on Sociodemographic variables (gender and age), Public Health Measures and COVID variables (quarantine status by city, status of the vaccination process, infected by COVID-19 and symptoms of COVID-19), Physical activity support variables (use of the sports time, type of housing, access to green areas and family income in Chilean pesos), and Educational variables (study programs, years of study and hours of study per day).

In addition, in section 2.2 Procedure and instruments, page 3 of the manuscript, the following sentence was added to the statement "mass vaccination process": "(2-dose vaccination schedule 21 days apart)". The foregoing, with the purpose of being able to specify more in the information provided and give consistency with what is presented in Table 1.

Comment 8. The study was based on PHI, GHQ-12, PHQ-9 as described in Section 2.2. It is strongly needed to justify the applicability of these measure items to the context being studied, i.e. young university students in Chile. It is quite difficult to judge how these were well accepted and used generally, e.g., appeared as [60, 61]. The list of items may be provided in the appendix for interested readers

Response 8: This study wanted to address the relationship between physical activity and psychological health. To do so, we decided to employ measures of well-being and mental health as complementary approaches. Regarding well-being, we decided on subjective well-being since it prioritized a hedonic perspective (instead of psychological wellbeing). As a measure, we opted to use the experienced well-being scale of the Pemberton Happiness Index (PHI), an instrument that evaluates the affective state in a limited time and it has been validated in Spanish and Latin-American countries including Chile, showing acceptable adjustment indices.

Hervás G, Vázquez C. Construction and validation of a measure of integrative well-being in seven languages: The Pemberton Happiness Index. Heal Qual Life Outcomes 2013 111 [Internet]. 2013 Apr 22 [cited 2021 Jul 31];11(1):1–13. Available from: 10.1186/1477-7525-11-66

Martínez-Zelaya G, Bilbao M, Costa D, da Costa S. Bienestar y su medida: validación del Pemberton Happiness Index en 3 países latinoamericanos. Rev Latinoam Psicol Posit / Lat J Posit Psychol [Internet]. 2018;4:125–40.

From the two PHI dimensions, it was decided to evaluate experienced well-being instead of remembered well-being since the latter is not consistent with the need to identify the variability of the emotional experience in the short term. Experienced well-being include items that present experiences with a positive affective tone (e.g. I learned something interesting, I felt satisfied with something I did, I did something that I really enjoy doing) and experiences with a negative affective tone (e.g. at times I felt overwhelmed, I was bored for a long time, I was worried about personal issues) to which people answer if they have experienced it or not.

Together with measures of well-being, we added measures of mental health by the General Health Questionnaire, a well-recognized instrument to assess general mental health, which has also been validated in Chilean samples:

Rivas-Diez, R., & del Pilar Sanchez-Lopez, M. (2014). Psychometric properties of the general health questionnaire (GHQ-12) in Chilean female population/Propiedades psicométricas del cuestionario de salud general (GHQ-12) en población femenina chilena. Revista Argentina de Clínica Psicológica, 23(III), 251.

Garmendia ML. Análisis factorial: una aplicación en el cuestionario de salud general de Goldberg, versión de 12 preguntas. Rev Chil Salud Pública [Internet]. 2007 Jan 1 [cited 2021 Jul 29];11(2):57–65.

Also, it has been validated in Spanish speaking samples:

Herrera, R. M., Calle, C. L., Ramirez, M. C. R., & Castro, J. L. (2018). Estructura factorial y fiabilidad del Cuestionario de Salud General de Goldberg (GHQ-12) en universitarios ecuatorianos. Revista Argentina de Ciencias del Comportamiento (RACC), 10(3), 35-42.

Lobos-Rivera, M. E., & Gutiérrez-Quintanilla, J. R. (2022). Adaptación psicométrica del Cuestionario de Salud General (GHQ-12) en una muestra de adultos salvadoreños.

It has been used in Chilean samples of university students:

Kobus, Valentina, María José Calletti, and Jaime Santander. "Prevalencia de síntomas depresivos, síntomas ansiosos y riesgo de suicidio en estudiantes de medicina de la Pontificia Universidad Católica de Chile." Revista chilena de neuro-psiquiatría 58, no. 4 (2020): 314-323.

Romero, M. I., Santander, J., Hitschfeld, M. J., Labbé, M., & Zamora, V. (2009). Consumo de tabaco y alcohol entre los estudiantes de medicina de la Pontificia Universidad Católica de Chile. Revista médica de Chile, 137(3), 361-368.

And in general Chilean population in the context of COVID:

Alcover, C. M., Salgado, S., Nazar, G., Ramirez-Vielma, R., & Gonzalez-Suhr, C. (2020). Job insecurity, financial threat, and mental health in the COVID-19 context: The buffering role of perceived social support. MedRxiv.

Castillo, H. B., González, G. Q., & Contreras, L. Á. (2021). Efecto programa de telerehabilitación sobre la salud mental y el estrés en pacientes sobrevivientes covid-19. Un studio piloto. Revista Chilena de Rehabilitación y Actividad Física, (1), 1-16.

Gutiérrez, R. J., Robles, E. M., & Álvarez, A. V. (2021). Salud Mental y Apoyo Social en habitantes de Copiapó, Chile, en el contexto de la COVID-19. Psicogente, 24(46), 1-16.

Regarding mental health symptoms some items from the Patient Health Questionnaire were selected specifically those that address 1. Sleeping disturbances, 2. feeling tired or lack of energy, 3. changes in eating behavior, and 4. concentration problems.

Its psychometric properties have been analyzed in the Chilean sample concluding that PHQ-9 is a useful instrument for the screening of depressive disorders:

Saldivia, S., Aslan, J., Cova, F., Vicente, B., Inostroza, C., & Rincón, P. (2019). Propiedades psicométricas del PHQ-9 (Patient Health Questionnaire) en centros de atención primaria de Chile. Revista médica de Chile, 147(1), 53-60.

PHQ also show validation in Chilean adolescents:

Borghero F, Martínez V, Zitko P, et al. [Screening depressive episodes in adolescents. Validation of the Patient Health Questionnaire-9 (PHQ-9)]. Revista Medica de Chile. 2018 Apr;146(4):479-486.

And rural samples:

Caneo, C., Toro, P., Ferreccio, C., Bambs, C., Cortés, S., Paredes, F.,... & Mauco Research Team. (2020). Validity and Performance of the Patient Health Questionnaire (PHQ-2) for Screening of Depression in a Rural Chilean Cohort. Community mental health journal, 56(7).

PHQ-9 has been used in Chilean university students: Álamo, C., Antúnez, Z., Baader, T., Kendall, J., Barrientos, M., & Barra, D. D. L. (2020). The sustained increase of mental health symptoms in Chilean university students over three years. Revista Latinoamericana de Psicología, 52, 71-80.

Comment 9. The results in Table 1 and 2 were not well explained in the paragraph. It is suggested to interpret and explain them in an orderly manner. Short conclusion will be useful as how it was judged from the results in Table 1 and 2 respectively. 

Response 9: We appreciate the reviewer's comment that allowed us to improve the wording of the results, which will make it easier for readers to understand. In this sense, we have followed the reviewer's suggestion and have added a brief conclusion on how the results of Table 1 and 2 were evaluated, respectively. Therefore, the third through fifth paragraphs on page 5 and the first and second paragraphs on page 6 were modified, corresponding to section 3 of the Results, which specifically describe the findings reported in Table 1 of the Manuscript, was as follows:

“Table 1 shows descriptive data on the characteristics of the participants according to sociodemographic variables, public health, and COVID measures, support for physical activity, and educational characteristics. Overall, university students were mostly women (66.0%), the majority reported an age between 18-20 years (34.8%). The students were mainly in a quarantined city only on weekends (61.5%), about a third of the students were inoculated with a first dose (29.6%) or a second dose of vaccination (37.4%), and mainly the students reported not having contracted the COVID-19 disease (88.0%). The majority never used the time band for sports (69.9%), lived in houses (82.5%), had access to green areas (88.2%) and had a family income in Chilean pesos of 296,000 - 607,000. Most of the participants were studying some university degree related to health area (46.3%), were in the third (22.8%) or between fifth and seventh year of the degree (22.8%), and most report-ed studying from 6 to 8 hours a day (29.8%).

Additionally, Table 1 shows the significant differences according to the descriptive data of the characteristics of the participants in variables of experienced well-being, positive and negative affective experience, and general mental health. Specifically, in the sociodemographic variables, it is observed that women present an average of significantly worse general mental health than men (17.77 ± 5.85 v/s 15.75 ± 6.67; t (380) = 9.242, p = 0.003, r2 = 0.42, with a small effect size). In public health measures and COVID variables, when symptoms due to COVID infection occurred, the group with mild symptoms had a significantly lower mean of positive affective experiences than the asymptomatic group (3.38 ± 1.25 v/s 4.44 ± 0.73; t (44) = 5.942, p = 0.019, r2 = 0.66, with a medium effect size).

Regarding the variables to support physical activity, those who sometimes used the sports time band had a mean of positive affective experiences significantly higher than that of the group that never used the sports time band (3.82 ± 1.14 v/s 3.13 ± 1.31; t (380) = 23.757, p < 0.001, r2 = 0.773, with a medium effect size), as well as significantly better aver-age general mental health than the group that never used the sports time band (15.90 ± 5.97 v/s 17.59 ± 6.25; t (380) = 6.019, p = 0.015, r2 = 0.295, with a small effect size). Similarly, those with access to green areas had significantly higher average experienced well-being than those without access to green areas (5.96 ± 1.67 v/s 5.00 ± 1.70; t (380) = 12.983, p < 0.001, r2 = 0.554, with a medium effect size), a significantly higher mean positive affective experience than those who did not have access to green areas (3.41 ± 1.25 v/s 2.82 ± 1.57; t (380) = 8.177, p = 0.004, r2 = 0.386, with a small effect size), and significantly better mean general mental health than those who did not have access to green areas (16.68 ± 6.08 v/s 20.11 ± 6.36; t (380) = 12,500, p < 0.001, r2 = 0.539, with a medium effect size) (Table 1).

Regarding educational variables, it was found that those who studied a program in social sciences or humanities had a significantly lower average of positive affective experiences than those who were pursuing degrees in the areas of education, engineering/management, or health (2.94 ± 1.24 v/s 3.84 ± 1.16 v/s 3.35 ± 1.37 v/s 3.34 ± 1.29; F (3,378) = 5.679, p = 0.001, η2 = 0.043, with a trivial effect size) (Table 1).).”

Likewise, we reformulate the way of writing the findings reported in Table 2. Therefore, the first three paragraphs on page 9 that refer to section 3 of the Manuscript Results and describe the results of Table 2, was as follows:

“Table 2 shows a non-adjusted multifactorial model (Model 0) and an adjusted one (Model 1). Both models analyze the significant differences and the confidence intervals of the main effects for the variables of experienced subjective well-being, positive and negative affective experience, and general mental health, from the behavioral profiles of physical activity and sedentary time, using the PI-S behavioral profile as the reference group.

For the non-adjusted multifactorial model (Model 0), behavioral profiles PA-NS (p <0.001), PA-S (p <0.001), and PI-NS (p <0.05) showed higher levels of experienced well-being and positive affect than university students who had a PI-S behavioral profile. In the case of negative affective experiences, no significant differences were reported between the behavioral profiles of physical activity and sedentary time. On the other hand, the university students with behavioral profiles PA-NS (p < 0.001), and PA-S (p < 0.01) showed better general mental health than the university students with the PI-S behavioral profile (Table 2).

For the adjusted multifactorial model (Model 1), the significant differences and confidence intervals of the main effects were analyzed from the cofactors (gender, symptoms of COVID-19, time band for sports, access to green areas and study program) detected in the initial analysis (Table 1). In this sense, university students with behavioral profiles PA-NS (p <0.001), PA-S (p <0.001), and PI-NS (p <0.05) showed higher levels of experienced well-being and positive affect than university students who had a PI-S behavioral profile. Additionally, university students with a PA-NS (p <0.001) showed with better general mental health than the university students with a PI-S behavioral profile (Table 2).”

Additionally, the order of presentation of the subjective well-being and general mental health variables in Tables 1 and 2 was modified, in order to generate consistency with what is described in the text. Fitted both tables as follows:

TOTAL

N (%)

 Experienced well-being

M (SD)

Positive affective experience

M (SD)

Negative affective experience

M (SD)

General mental health

 M (SD)

Sociodemographic variables

Gender

Women

Men

252 (66.0%)

130 (34.0%)

5.81 (1,72)

5.92 (1,67)

3.27(1,29)

3.47 (1.31)

2.46(1.46)

2.55(1.59)

17.77 (5.85)

  15.75 (6.67) **

Age

18 – 20 years

21 – 22 years

23 – 29 years

133 (34.8%)

117 (30.6%)

132 (34.6%)

6.06 (1.79)

5.59 (1.53)

5.86 (1.74)

3.44 (1.27)

3.34 (1.29)

3.23 (1.33)

2.38(1.61)

2.75(1.45)

2.37(1.41)

16.59 (6.08)

17.84 (6.15)

16.91 (6.35)

Public health measures and COVID variables

Quarantine status per city

Total quarantine     

Weekend quarantine     

No quarantine     

103 (27.0%)

235 (61.5%)

44 (11.5%)

5.92 (1.60)

5.87 (1,76)

5.55 (1.64)

3.59 (1.18)

3.23 (1.34)

3.30 (1.25)

2.67 (1.55)

2.37 (1.48)

2.75 (1.48)

17.20 (6.15)

16.89 (6.38)

17.82 (5.44)

Vaccination process status

Non-vaccinated

One-dose vaccinated

Two-dose vaccinated

126 (33.0%)

113 (29.6%)

143 (37.4%)

5.64 (1.71)

5.80 (1.44)

6.06 (1.86)

3.33 (1.40)

3.37 (1.27)

3.32 (1.24)

2.68 (1.57)

2.58 (1.51)

2.26 (1.42)

17.56 (6.02)

17.54 (6.28)

16.31 (6.27)

Infected by COVID-19

Non-infected

infected

336 (88.0%)

46 (12.0%)

5.86 (1.70)

5.76 (1.78)

3.30 (1.30)

3.59 (1.24)

2.45 (1.50)

2.83 (1.54)

17.23 (6.31)

16.02 (5.30)

Symptoms of COVID-19

Asymptomatic

Mild symptoms

9 (19.5%)

37 (80.5%)

6.33 (1.32)

5.68 (1.19)

4.44 (0.73)

  3.38 (1.25) *

3.11 (1.96)

2.70 (1.50)

13.56 (4.40)

16.43 (5.27)

Physical activity support variables

Use of the sports time band

Never used

Sometimes used

267 (69.9%)

115 (30.1%)

5.76 (1.81)

6.05 (1.41)

3.13 (1.31)

   3.82 (1.14) ***

2.37 (1.48)

2.77 (1.54)

17.59 (6.25)

  15.90 (5.97) *

Type of home

House

Apartment

Countryside house

315 (82.5%)

33 (8.6%)

34 (8.9%)

5.87 (1.71)

5.94 (1.73)

5.50 (1.62)

3.34 (1.31)

3.12 (1.14)

3.53 (1.37)

2.47 (1.52)

2.18 (1.46)

3.03 (1.24)

17.08 (6.26)

17.33 (6.72)

16.88 (5.27)

Green areas access

No green areas access

green areas access

45 (11.8%)

337 (88.2%)

5.00 (1.70)

   5.96 (1.67) ***

2.82 (1.57)

   3.41 (1.25) ***

2.82 (1.46)

2.45 (1.50)

20.11 (6.36)

   16.68 (6.08) ***

Family income in Chilean pesos (CLP)

< 296.000

296.000 - 607.000

608.000 - 1.572.999

> 1.573.000

56 (14.7%)

145 (38.0%)

124 (32.4%)

57 (14.9%)

5.59 (1.74)

5.83 (1.74)

5.81 (1.72)

6.19 (1.53)

3.09 (1.35)

3.41 (1.32)

3.34 (1.24)

3.40 (1.32)

2.50 (1.39)

2.57 (1.49)

2.52 (1.50)

2.21(1.68)

18.16 (5.96)

17.32 (5.88)

16.90 (6.65)

15.81 (6.17)

                                                                                               Educational variables

Study programs

Education

61 (16%)

6.11 (1.67)

3.84 (1.16)

2.72 (1.63)

17.00 (6.67)

Engineering/Management

66 (17.3%)

5.53 (1.68)

3.35 (1.37)

2.82 (1.56)

16.92 (6.16)

Health area

177 (46.3%)

5.98 (1.79)

3.34 (1.29)

2.36 (1.48)

17.08 (6.17)

Social Sc./Humanities

78 (20.4%)

5.59 (1.63)

   2.94 (1.24) **

2.35 (1.35)

17.29 (6.06)

Study years

1

2

3

4

5 to 7

57 (14.9%)

73 (19.1%)

87 (22.8%)

78 (20.4%)

87 (22.8%)

5.75 (1.79)

5.74 (1.60)

6.05 (1.78)

5.71 (1.59)

5.92 (1.76)

3.26 (1.39)

3.41 (1.22)

3.34 (1.27)

3.47 (1.21)

3.20 (1.40)

2.51 (1.71)

2.67 (1.47)

2.30 (1.55)

2.77 (1.43)

2.28 (1.37)

17.19 (5.37)

17.53 (6.43)

17.15 (6.33)

17.35 (5.93)

16.33 (6.69)

Study hours per day

1-3

4-5

6-8

9-12

111 (29.1%)

91 (23.8%)

114 (29.8%)

66 (17.3%)

5.68 (1.69)

5.99 (1.81)

5.98 (1.59)

5.70 (1.75)

3.18 (1.31)

3.45 (1.34)

3.35 (1.19)

3.42 (1.40)

2.50 (1.58)

2.46 (1.56)

2.37 (1.45)

2.73 (1.39)

17.41 (6.25)

17.15 (6.11)

16.72 (5.87)

17.08 (6.89)

Table 2. Estimated effect of physical activity behavioral profile and sedentary time.

Model 0

Non-adjusted

Model 1

Adjusted

Variables

βáµ¢ [CI 95%]

βáµ¢ [CI 95%]

Experienced well-being

PA-NS

1.14 [0.65; 1.62]***

1.08 [4.48; 5.73]***

PA-S

0.88 [0.46; 1.29]***

0.86 [0.44; 1.28]***

PI-NS

0.87 [0.18; 1.55]*

0.73 [0.05; 1.42]*

PI-S

Reference

Reference

Positive affective experience

PA-NS

1.08 [0.72; 1.45]***

0.84 [0.47; 1.20]***

PA-S

0.75 [0.44; 1.06]***

0.64 [0.33; 0.95]***

PI-NS

0.63 [0.12; 1.15]*

0.52 [0.02; 1.02]*

PI-S

Reference

Reference

Negative affective experience

PA-NS

-0.054 [-0.50; 0,39]

-0.23 [-0.70; 0.22]

PA-S

-0.124 [-0.50; 0.25]

-0.22 [-0.61; 0.16]

PI-NS

-0.232 [-0.86; 0.40]

-0.21 [-0.84; 0.41]

PI-S

Reference

Reference

General mental health

PA-NS

-3.67 [-5.46; -1.88]***

-2.84 [-4.7; -0.98]**

PA-S

-2.02 [-3.54; -0.50]**

-1.41 [-2.97; 0.15]

PI-NS

-2.36 [-4.9; 0.18]

-2.24 [-4.76; 0.28]

PI-S

Reference

Reference

Comment 10. Presentation of the content in Figure 1 and Table 2 should be consistent. For example, well-being is presented as PHI in Figure 1a but as Experience well-being in Table 2.

Response 10: Based on the reviewer's comment 10, we made important changes to Figure 1 of the manuscript, with the purpose of making what was presented consistent with the methodology section. In this way, the name of the variables was replaced by experienced well-being, positive affective experiences, negative affective experiences, general mental health. The order of the graphs was changed, for greater consistency in the way the results were delivered in Tables 1 and 2, and with the methodology section. The x-axis score of each graph was modified from the maximum score of each variable, that is, 10 points for experienced well-being, 5 points for positive and negative affective experiences, and 36 points for general mental health. Additionally, and solely for the purpose of completing the information provided, a "cut-off score" was added, based on evidence, for the variables of experienced well-being and general mental health. The foregoing was explained and detailed both in the legend of Figure 1, and in section 2.2 Procedure and instruments. Next, we present Figure 1 with the modifications already mentioned.

Figure 1. Subject well-being and general mental health variables according to the physical activity behavioral profile and sedentary time.

Note. PA-NS = physically active and non-sedentary; n = 84, PA-S = physically active and sedentary; n = 172, PI-NS = physically inactive and non-sedentary; n = 29 and PI-S = physically inactive and sedentary; n = 97. Scores of six or less indicate low experienced well-being, and scores of seven or more indicate high experienced well-being. The general mental health variable on the GHQ-12 scale, the higher the score, the worse general mental health, where scores equal to or less than sixteen can be considered normal and scores greater than sixteen can be considered high. Variables are presented as mean and their respective 95% confidence interval.

Similarly, the variables were renamed both in Table 2 and in the text that describes the pre-established findings in the table, being replaced by experienced well-being, positive and negative affective experiences, and general mental health. For more detail, you can see the response to comment 9 where these inconsistencies presented in Table 2 were resolved.

Comment 11. Figure 1, what is F(3;591)? What is 591? 

Response 11: We appreciate the reviewer's comment that allowed us to correct the values of the statistical test presented in Figure 1. Where it said F(3591), it now says F(3379). For more details on the changes to Figure 1, see the response to reviewer comment 10

Comment 12. PANAS appears in Table 2 without proper definition.

Response 12: Regarding Figure 1, we agree with the reviewer's comment that the abbreviation PANAS does create confusion. In addition, to respond to comment 9 of the reviewer, and in order to generate a logical connection between what is mentioned in the methodology section, as well as the results of the manuscript. The abbreviation PANAS in Table 2 was replaced by positive and negative affective experience.

Comment 13. Explanation of Table 2 is too coarse.

Response 13: We agree with the reviewer that the explanation for the findings reported in Table 2 is very coarse. Similarly, we appreciate these comments made by the reviewer, which allowed us to strengthen the writing of the results section of our manuscript. In this sense, previously in comment 9, the need to improve the explanation of the results referring to Tables 1 and 2 was suggested. Therefore, as noted in response to comment 9, modified the paragraphs on page 9 that refer to section 3 of the Results of the manuscript and describe the results of Table 2. With the purpose of detailing that Table 2 shows an unadjusted multifactorial model (Model 0) and an adjusted one (Model 1). And that both models analyze the significant differences and the confidence intervals of the main effects for the subjective well-being and general mental health variables, using the PI-S behavioral profile as the reference group.

Comment 14. Section 4 (around lines 277 to 315) sounds more like a literature review than discussion. 

Response 14: We appreciate the comment made by the reviewer. In this sense, based on the suggestions made in this and other comments, we have made changes that have substantially improved the Discussion section. Likewise, we believe that the evidence presented in this section allows for a contrast with the existing information regarding physical activity and sedentary lifestyle and its association with well-being and mental health in university students during the COVID-19 pandemic.

Comment 15. In my opinion, line 320, it sounds too ambitious to claim the scientific strength of the study in the area (supposed to be in Latin America, or specifically Chile). It has been present in other studies but just in other geographical areas. 

Response 15: We agree with the reviewer that the way the sentence was worded expresses an overly ambitious statement. Therefore, we have modified the sentence that begins the fifth paragraph of the Discussion. In this way, we emphasize that our findings are a contribution to the evidence of the effect that a sedentary lifestyle has on mental health, as well as that these results correspond to Chilean university students. The sentence indicated by the reviewer now reads like this:

“However, despite the aforementioned studies on university students investigated in the COVID-19 pandemic on physical activity and mental health, our study, carried out in Chilean university students with a comparative design of four groups (behavioral profiles), provides findings that support the effect of sedentary lifestyle as one of the variables that most affect mental health

Comment 16. Chi-squared should be the Greek alphabet rather than X in the paragraph above Table 4.

Response 16: We have changed the symbol to identify the chi-square test, to the Greek alphabet symbol “χ2”, throughout the paragraph before Table 3

Comment 17. Line 339 to 341, 367 to 370, and 383 to 385 overly repeat the preceding sections. 

Response 17: Based on comment 17 made by the reviewer, we have revised the strengths and limitations section of the manuscript. To eliminate inconsistencies and repetitive sentences. In this way, we have made modifications and given a new order to the paragraphs that make up the Strengths and limitations section. It was as follows:

“This study is the first to analyze the behavioral profiles of physical activity and sedentary time with subjective well-being and mental health in university students from different universities and regions of Chile during the COVID-19 pandemic. In this sense, the present study is a contribution to the knowledge about how the behavioral profile of physical activity and sedentary lifestyle can differentially modulate well-being and mental health. In addition, this study allowed generating a pattern of subjective well-being and mental health according to the different behaviors of physical activity and sedentary lifestyle of university students, which could help understand which groups of students have the worst indicators of well-being and mental health. Finally, this study contributes to understanding the role of physical activity and sedentary lifestyle in well-being and mental health, providing theoretical and empirical support to generate programs to promote physical activity and reduce sedentary lifestyle in the university population.

But nevertheless, the present study is not without its limitations. First, due to the public health measures derived from the pandemic, the variables studied were self-reported through the internet, specifically self-reported physical activity is subject to overestimation compared to objective measures such as accelerometry [81]. Second, the cross-sectional design does not allow determining causal inferences about relationships between variables. Third, due to the convenience sampling method, selection bias is also a possibility. Fourth, even though this study considers different sociodemographic variables, well-being and mental health are constructs that can be modulated by multiple factors that were not measured in this study, such as family and academic relationships. Finally, this study does not distinguish between planned and unplanned (incidental) physical activity, given the motivational and behavioral differences between these types of physical activity, it is plausible that they are associated with differences in well-being and mental health [46].

One of the future lines of research, considering the design of this study, is to delve in-to the effects on well-being and mental health that the behavioral profile of physical activity and sedentary lifestyle could have in different populations with or without pathologies of physical and psychological health, as well as in different age groups. Similarly, it is convenient to continue studying the relationship between these variables but using objective measurement instruments that complement the information provided by the self-report questionnaires [43,81]. For example, measures of facial and non-verbal expression in the case of well-being [43], and for physical activity the use of accelerometers, which would allow a more precise recording of planned and unplanned physical activity [81]. Another aspect is the study of how much the different ways of coping with confinement may have affected the behavioral profile of active activity and sedentary lifestyle of university students.”

Comment 18. Conclusions in Section 5 are superficial and not emphasizing the research contribution, therefore need a major revise.

Response 18: Based on the reviewer's comment, we have made modifications to the conclusion stated in the manuscript. With the purpose of highlighting the analysis carried out in this study, in terms of the findings and contributions provided by the analysis of behavioral profiles that integrate activity and sedentary time. In this way, the Conclusion section of the manuscript is as follows:

The subjective well-being and mental health of Chilean university students was affected differently according to the profile of physical activity and sedentary time that the students presented during the COVID-19 pandemic. In this line, this study is the first to make a comparison of this type of behavior profiles in university students in the context of COVID-19 pandemic. Out of the four behavioral profiles, university students who present-ed a physically inactive and sedentary one had worse experienced well-being, positive affective experiences, and general mental health. Additionally, students who presented a physically active and non-sedentary profile were associated with better general mental health than those who were physically active and sedentary. In this area, increasing physical activity and reducing sedentary lifestyle can promote well-being and mental health in the university student population during the COVID-19 pandemic. In particular, within the behavioral profiles, a sedentary lifestyle is one of the variables that most affect the mental health of Chilean university students. While physical activity is beneficial for mental health, as reported in the literature and in the present study. Therefore, awareness on the harmful effects that a sedentary lifestyle causes on the mental health of young university students is essential. Thus, in times of a health crisis, government and university leaders must implement measures to encourage this population to maintain adequate levels of physical activity, as well as reduce sedentary times.”

Comment 19. Generally speaking, the manuscript must be thoroughly revised to be more concise and scientifically rigorous. 

Response 19: Based on the important comments made by the reviewer, which allowed deepening the concepts presented in the introduction section, strengthening the methodology and writing of the results, as well as deepening the discussion of the manuscript. We have given greater consistency and scientific rigor to our proposal. We are very grateful for these suggestions and inputs.

Reviewer 2 Report

The authors presented an interesting manuscript in which they explored the relationships among physical activity and sedentary time behavioral profiles with well-being, general mental health, positive and negative affective experiences, and a sampling of mental health symptoms in Chilean university students during the COVID-19 pandemic. This is an important and timely topic given the mental health crisis affecting young adults, a crisis that has worsened during the pandemic. The authors bring up important points to consider and make a solid argument for the study. Inconsistent alignment of variable names throughout the manuscript made it difficult to follow at times. Also, be careful to be clear what conclusions follow directly from the statistical results versus what is conjecture or speculation that needs further investigation.

Manuscript suggestions:

Line 46: Clarify what is meant by “restrictions on movement”

Line 49: Delete “respond to”

Line 65 – 67: Please revise sentence for clarity. Point of sentence is unclear.

Hypotheses statements: Choose stronger language for your hypotheses. For example,

"We hypothesized that physically inactive and sedentary students would present with worse levels of subjective well-being and mental health than physically active and non-sedentary students (hypothesis 1)."

Line 105: The term correlational-causal is confusing. I would suggest using "correlational"

Line 115: Please revise the exclusion criterion #3. The exclusion criterion was probably "having reported a health and/or physical condition that prevented the engagement in physical activity during the previous six months." The way it is currently written states that having reported no such condition would be an exclusion. This seems unlikely to be an exclusion criterion.

Line 130: Revision would make this more clear. For example, "Physical activity and sedentary time were measured with the International ... "

Line 132-142: This makes it sound like it was one question asking about physical activity duration for each intensity level, but the questionnaire really asks about the frequency of PA (days per week) and then the duration (hours or minutes spent). Did you then multiply: Frequency X Duration X MET value? Consider revising lines 132 to 134 for a clearer explanation of the IPAQ-SF.

Lines 143 – 148: Since the IPAQ asks about light PA but the recommendations often focus on MPA and VPA only, can you clarify whether your calculation of MET/min/week for the categorization of physically active included light PA? Perhaps stating the international standards that were used would clarify this.

Lines 154 - 156. Please define what the "sports time band" was.

Lines 157 – 166: Was the PHI divided into subscales for positive and negative affective experiences? If so, please state it more clearly that the subscales were used as a part of the total as well as analyzed individually.

Lines 175 – 181: Since the PHQ-9 is commonly used in its entirety as a scale for depression, please revise this section to make it more clear that you are not attempting to measure depression as a mental health symptom but rather you have selected individual items from the PHQ-9 scale to measure: (1) fatigue ("..."), (2) Sleeping difficulty ("..."), (3) Eating behaviors ("..."), etc.  using the same variable names that you used in the results section including the tables.

Line 183: Was this sex or did you mean gender here? Tables label variable as gender.

Lines 182 – 185: Table 1 has these variables divided into large categories. It would be more clear if these categories were introduced here.

Lines 188-191: Sentence is unclear. Please revise.

Measures/Results and Tables: Use the same variable names in the measures section, results narrative, and tables. For example, Table 1 uses “PHI”, Table 2 states “Experienced well-being” and the measures section called it “subjective well-being”. Similar problems of alignment exist for the other variables.

GHQ-12: Several times throughout the manuscript this scale is interpreted as “deterioration of mental health” but deterioration suggests a decline over time. It is more accurate to state poor mental health when interpreting data from a cross-sectional collection of data using this scale. If this questionnaire asks about a change of symptoms over time, please add this information to the measures section.

Table 2: Remove PANAS as variable label. PANAS usually refers to the Positive and Negative Affect Schedule by Watson et al. (1988).

Lines 281 – 292: Statement of findings goes beyond what the statistical analyses support. Revise and be clear as to what the results truly support versus what is speculation or conjecture, and thus needs further testing in future studies.

Lines 323 – 326: Unclear statement. Make sure not to extrapolate beyond the results of the analyses unless it is clear conjecture and preparation for future exploration.

Lines 361-362: Please include a reference for the use of measures of facial and non-verbal expression for well-being.

Lines 372 - 376: Sentence is unclear. There may be punctuation missing.

Author Response

The authors presented an interesting manuscript in which they explored the relationships among physical activity and sedentary time behavioral profiles with well-being, general mental health, positive and negative affective experiences, and a sampling of mental health symptoms in Chilean university students during the COVID-19 pandemic. This is an important and timely topic given the mental health crisis affecting young adults, a crisis that has worsened during the pandemic. The authors bring up important points to consider and make a solid argument for the study. Inconsistent alignment of variable names throughout the manuscript made it difficult to follow at times. Also, be careful to be clear what conclusions follow directly from the statistical results versus what is conjecture or speculation that needs further investigation.

Manuscript suggestions:

Comment 1. Line 46: Clarify what is meant by “restrictions on movement”

Response 1: We value the reviewer's comment, which allowed us to clarify the concepts mentioned in the manuscript, related to the mitigation measures that were carried out in Chile and that sought to prevent the spread of the pandemic.

Therefore, in the introduction to the manuscript, specifically in the second paragraph, where the public health measures implemented by the Government of Chile are mentioned, we have decided to deepen and specify the concept of “restrictions on movement”. The second paragraph, page 2 of section 1. Introduction, now reads as follows:

Specifically, the Chilean government implemented public health measures such as restrictions on movement between cities and access to public places (fitness centers, cinemas, restaurant, among others), social distancing, and dynamic periods of quarantine (i.e., while some cities could go into total or weekend quarantine, others left that state, depending on the local epidemiological situation) [11].

Comment 2. Line 49: Delete “respond to”

Response 2: To respond to the reviewer's comment, we have removed the "respond to" statement from the first paragraph on page 2.

Comment 3. Line 65 – 67: Please revise sentence for clarity. Point of sentence is unclear.

Response 3: We appreciate the reviewer's comment that allowed us to delve into the concept we gave in the introduction of "sedentary lifestyle", as well as the definition we gave of the "physically active but sedentary" paradox. Consequently, the fourth paragraph on page 2 referring to the introduction of the manuscript, is as follows

“On the other hand, those who spend more than four hours a day on behaviors such as driving, sitting, lying down, or spending time in front of a screen can be classified as sedentary [27,28]. In this regard, being physically active and sedentary are not mutually exclusive opposites; On the contrary, as a consequence of current lifestyles, it is possible to comply with the recommendations of minimal physical activity per week, and also present sedentary times greater than four hours per day, which is known as the “physically active but sedentary” paradox [29].”

Comment 4. Hypotheses statements: Choose stronger language for your hypotheses. For example, "We hypothesized that physically inactive and sedentary students would present with worse levels of subjective well-being and mental health than physically active and non-sedentary students (hypothesis 1)."

Response 4: We appreciate the suggestions made in comment 8 by the reviewer. In this way, the first paragraph of page 3 of the manuscript, the sentence referring to hypothesis 1 that we propose, is as follows:

“We hypothesize that physically inactive and sedentary students will have worse levels of subjective well-being and mental health than physically active and non-sedentary students (hypothesis 1).”

Comment 5. Line 105: The term correlational-causal is confusing. I would suggest using "correlational"

Response 5: We agree with the reviewer's comment that the term 'correlational-causal' can cause some confusion. However, we believe that it is pertinent to maintain the concept of a causal-correlational study, given the characteristics of how the data analysis was carried out in this study. Since, not only descriptive and association analyzes were performed. In addition, multifactorial models were made, and the use of dependent and independent variables, with the purpose of explaining the causes of subjective well-being and mental health, based on the profile of physical activity and sedentary time. This concept is neither new nor less created by the authors, but rather it is a subclassification of correlational design studies, in which an attempt is made to explain the phenomenon to be studied (Hernández Sampieri et al., 2014).

Hernández Sampieri, R., Fernández Collado, C. y Baptista Lucio, P. (2014). Metodología de la investigación. México: Editorial Mc Graw Hil.

Comment 6. Line 115: Please revise the exclusion criterion #3. The exclusion criterion was probably "having reported a health and/or physical condition that prevented the engagement in physical activity during the previous six months." The way it is currently written states that having reported no such condition would be an exclusion. This seems unlikely to be an exclusion criterion.

Response 6: We agreed that the wording of exclusion criterion 3 was unclear and confusing. Therefore, and as the reviewer suggests in comment 6, the definition of exclusion criterion 3, second paragraph of section 2.1 Design and participants, is as follows:

“(3) having reported no health and/or physical condition that prevented the engagement in physical activity during the previous six months”

Comment 7. Line 130: Revision would make this more clear. For example, "Physical activity and sedentary time were measured with the International ... "

Response 7: We have revised the sentence indicated by the reviewer in comment 8. Y to be consistent with the way instruments are described in section 2.2. Procedure and instruments of the manuscript. The first paragraph referring to "Level of physical activity and sedentary time" of section 2.2 Procedure and instruments was modified, being as follows:

“The International Physical Activity Questionnaire (IPAQ) was used in its short version of seven items, which consists of a recall measure of seven days”

Comment 8. Line 132-142: This makes it sound like it was one question asking about physical activity duration for each intensity level, but the questionnaire really asks about the frequency of PA (days per week) and then the duration (hours or minutes spent). Did you then multiply: Frequency X Duration X MET value? Consider revising lines 132 to 134 for a clearer explanation of the IPAQ-SF.

Response 8: We appreciate the reviewer's comment that allows us to clarify the procedure that led us to determine the level of physical activity of the participating students. Based on reviewer comments and suggestions, we have improved the wording to better explain how the required information was administered and extracted from the IPAQ self-report questionnaire. From these modifications, it is better understood that participants reported frequency and duration for each type of physical activity (intensity). And that the METS for each intensity were calculated from the multiplication of the MET value, the frequency and the duration of the physical activity performed. In this way, the first paragraph referring to "Level of physical activity and sedentary time" of section 2.2 Procedure and instruments was modified, being as follows:

“For the level of physical activity, participants must report the frequency (days per week) and duration (hours and minutes) of vigorous, moderate and light physical activity performed during the previous week. The level of physical activity is expressed through the metabolic equivalent (MET) minutes per week, which corresponds to the sum of the METs of physical activity of light, moderate and vigorous intensity. The METs for each intensity are obtained by multiplying the MET value (3.3 METs for light intensity, 4.0 METs for moderate intensity and 8.0 METs for vigorous intensity) by the total minutes per week of each type of intensity of physical activity [58]”

Comment 9. Lines 143 – 148: Since the IPAQ asks about light PA but the recommendations often focus on MPA and VPA only, can you clarify whether your calculation of MET/min/week for the categorization of physically active included light PA? Perhaps stating the international standards that were used would clarify this.

Response 9: To address reviewer comment 9 and based on previous comments. We have substantially modified the paragraphs that describe the process of applying and extracting data from the IPAQ questionnaire. In this way, we believe to answer and clarify the process that led us to use the commonly used categorization of physically inactive and active, based on the categories of low, moderate or high physical activity, provided by the IPAQ questionnaire. Additionally, we decided to remove the statement on international physical activity guidelines, as we agree that they can cause confusion. In addition, updated references have been added to support the procedures performed with the IPAQ questionnaire (attached below the modified text).

“For the level of physical activity, participants must report the frequency (days per week) and duration (hours and minutes) of vigorous, moderate and light physical activity per-formed during the previous week [58]. The level of physical activity is classified into three levels (low, moderate, and high), based on the total metabolic equivalents (METs) per week, whose value corresponds to the sum of the METs of physical activity of light, moderate and vigorous intensity [58]. The METs for each intensity are obtained by multiplying the MET value (3.3 METs for light intensity, 4.0 METs for moderate intensity and 8.0 METs for vigorous intensity) by the total minutes per week of each type of intensity of physical activity [58,59]. For sedentary time, participants were asked to report the hours and minutes spent sitting during a weekday (for example, in a class, at home, during free time, on the bus, watching television, etc.) [58]. This self-report questionnaire, validated and recommended to evaluate physical activity [57], has been previously applied in the university population in Chile [60].

Based on the surveyed students’ self-reports of physical activity level and sedentary time, a profile of physical activity behavior was created. Those who reported a low level of physical activity were considered physically inactive, while those who reported a moderate or high level of physical activity were considered physically active [58]. Also, it was classified as sedentary (sedentary time of > 4 hours per day) or non-sedentary (sedentary time of ≤ 4 hours per day), according to the report of hours of sedentary behavior per day [27,28]. Based on these criteria, the participants were classified into four groups: (1) physically active and non-sedentary (PA-NS), (2) physically active and sedentary (PA-S), (3) physically inactive and non-sedentary (PI-NS), and (4) physically inactive and sedentary (PI-S).”

  1. Ainsworth, B. E., Haskell, W. L., Herrmann, S. D., Meckes, N., Bassett, D. R., Tudor-Locke, C., Greer, J. L., Vezina, J., Whitt-Glover, M. C., & Leon, A. S. (2011). 2011 Compendium of Physical Activities: a second update of codes and MET values. Medicine and Science in Sports and Exercise, 43(8), 1575–1581. https://doi.org/10.1249/MSS.0B013E31821ECE12
  2. International Physical Activity Questionnaire. (2005). Guidelines for Data Processing and Analysis of the International Physical Activity Questionnaire (IPAQ) – Short and Long Forms. International Physical Activity Questionnaire.

Comment 10. Lines 154 - 156. Please define what the "sports time band" was.

Response 10: We appreciate the reviewer's comment, which allowed us to better explain the concept of "sports time band". This measure that was implemented by the Government of Chile, in response to the difficulty in carrying out physical activity because of the initial isolation measures and movement restrictions applied during the first waves of the covid-19 pandemic (from early 2020 to mid 2021). For this reason, in the first paragraph of section 2.1 Design and participants, the following definition was added that complements the concept that we indicated in the writing as "sports time band ". Now, after “sports time band”, it says:

(i.e., hours from 5 to 9 am for individual outdoor physical activity, for cities that were in total or weekend quarantine [11])”.

Comment 11. Lines 157 – 166: Was the PHI divided into subscales for positive and negative affective experiences? If so, please state it more clearly that the subscales were used as a part of the total as well as analyzed individually.

Response 11: We appreciate the reviewer's comment, which allowed us to delve into the process we carried out to analyze the well-being experienced, the positive and negative affective experiences, through the self-report delivered by the PHI. Thus, in section 2.2 Procedure and instruments, page 4. The second paragraph referring to Subjective Well-being was modified, thus remaining:

“For experienced well-being, the total score of the PHI scale was considered, where the items are converted into a single score from zero (zero positive experiences and five negative experiences) to 10 (five positive experiences and no negative experiences). Scores of six or less indicate low experienced well-being, and scores of seven or more indi-cate high experienced well-being [61]. The dimensions of positive affective experiences (5 items) and negative affective experiences (5 items) were also analyzed individually. For the positive affective experiences score, only the 5 items referring to this dimension of the PHI self-report questionnaire were considered, therefore, the score could range between 0 (zero positive experiences) and 5 (five positive experiences). Likewise, for the negative affective experience, only the 5 items referring to this dimension of the PHI self-report questionnaire were considered, therefore, the score could range between 0 (zero negative experiences) to 5 (five negative experiences).”

Comment 12. Lines 175 – 181: Since the PHQ-9 is commonly used in its entirety as a scale for depression, please revise this section to make it more clear that you are not attempting to measure depression as a mental health symptom but rather you have selected individual items from the PHQ-9 scale to measure: (1) fatigue ("..."), (2) Sleeping difficulty ("..."), (3) Eating behaviors ("..."), etc.  using the same variable names that you used in the results section including the tables.

Response 12: As the reviewer very well mentions, and in order to clarify that the 4 items of the PHQ-9 scale used were worked individually, and not with the purpose of detecting depression systems. We have made important changes in the paragraph referring to the study variable "mental health symptoms". Therefore, the paragraph of section 2.2 of procedure and instruments of the manuscript, referring to the mental health symptoms variable, becomes the following:

“Mental health symptoms: Four items were selected from the Patient Health Questionnaire (PHQ-9) [68], of the instrument adapted to Spanish by Barrigón et al. [69]. The four items selected to measure mental health symptoms were: (1) Sleeping problems (“Problems for falling asleep, staying asleep or sleeping too much”), (2) Fatigue (“Feeling tired or having little energy”), (3) Changes in eating behavior (“Poor appetite or eating too much”), and (4) Concentration problems (“Problems concentrating on something, like reading the newspaper or watching television”). This instrument con-siders the frequency of a personal situation in the previous week, with a 4-level Likert scale response format (1 = never to 4 = almost every day).”

Comment 13. Line 183: Was this sex or did you mean gender here? Tables label variable as gender.

Response 13: As the reviewer correctly mentions in comment 13, we were referring to the gender of the participating university students. Therefore, on page 5, specifically in the sociodemographic information paragraph, the word sex was replaced by gender. In this way, it is consistent with what is mentioned in Table 1 and throughout the manuscript.

Comment 14: Lines 182 – 185: Table 1 has these variables divided into large categories. It would be more clear if these categories were introduced here.

Response 14: Based on comment 14 of the reviewer, the paragraph belonging to section 2.2 Procedure and instruments, referring to sociodemographic information, is modified. So the first paragraph on page 5 looks like this

“Sociodemographic information: Sociodemographic information: An ad hoc questionnaire was developed, which collected information on Sociodemographic variables (gender and age), Public Health Measures and COVID variables (quarantine status by city, status of the vaccination process, infected by COVID-19 and symptoms of COVID-19), physical activity support variables (use of the sports time band, type of housing, access to green areas and family income in Chilean pesos), and educational variables (study programs, years of study and hours of study per day).

Comment 15. Lines 188-191: Sentence is unclear. Please revise.

Response 15: To respond to comments made by the other reviewer, we have substantially modified section 2.3 Statistical analysis. Because of this, we have removed the sentence noted in comment 15. The statistical analysis section now reads like this:

“Upon data collection on the different variables of the study: physical activity level and sedentary time, subjective well-being, mental health, and sociodemographic information, a descriptive statistical analysis is presented first. The qualitative data were represented by frequency and percentage, while the quantitative data by the mean and standard deviation. Data distribution was established by means of normality and equality variance tests (Shapiro-Wilk and Levene). The difference in means between two different groups was tested with the independent samples T-Student test. To determine the effect of profile of physical activity behavioral and sedentary time on the participants’ characteristics and subjective well-bing and general mental health were determined using one-way ANOVA. (*** p-value <0.001; ** p-value <0.01; * p-value <0.05). Cohen’s d effect size (ES) was calculated and qualitatively assessed as trivial (0–0.19), small (0.20–0.49), medium (0.50–0.79), or large (0.80 and greater) (Table 1, Figure 1, and Table 2).

Also, an adjusted multifactorial model was used to analyze the significance estimates of the cofactors main effects detected in the initial analysis. Data were presented as mean and its 95% CI. All analyses were incrementally adjusted according to different con-founding factors. Model 0 was unadjusted; Model 1 was adjusted for gender, symptoms of COVID-19, sports time band, access to green areas and study program (Table 1).

For both the unifactorial model and the multifactorial model, the following contrasts were applied: Bonferroni, to determine differences between the means and the fixed main effects, due to the four treatments or levels of the factor (p value <0.05).

 The Chi-square test (χ2) was used to establish the association between mental health symptoms and behavior profile (Table 3). All the clean data were statistically treated and submitted to the respective analysis using the SPSS Statistic 27 (2020) software. Significance at the level of p < 0.05 was used.”

Comment 16. Measures/Results and Tables: Use the same variable names in the measures section, results narrative, and tables. For example, Table 1 uses “PHI”, Table 2 states “Experienced well-being” and the measures section called it “subjective well-being”. Similar problems of alignment exist for the other variables.

Response 16: We appreciate the important comment made by the reviewer. By responding to this comment we were able to provide greater consistency between the procedures and instruments sections and the results reported in the manuscript. Thus, for subjective well-being, the variables of experienced well-being, positive affective experience and negative affective experience were considered. And for mental health, the variables of general mental health and mental health symptoms were considered. In this way, the manuscript was modified and unified by adding only the name of these variables, based on their description in section 2.2 of procedure and instruments, to later be reported in the Tables and Figures, as well as described in the text of the results section.

Comment 17. GHQ-12: Several times throughout the manuscript this scale is interpreted as “deterioration of mental health” but deterioration suggests a decline over time. It is more accurate to state poor mental health when interpreting data from a cross-sectional collection of data using this scale. If this questionnaire asks about a change of symptoms over time, please add this information to the measures section.

Response 17: We fully agree with the reviewer that it is more accurate to indicate poor mental health when interpreting data from a cross-sectional data collection using this scale. Therefore, the sentences where “mental health deficiency” is mentioned were modified.

Description of Findings Table 1, second paragraph of the Results section, page 5, now reads as follows:

“Specifically, in the sociodemographic variables, it is observed that women have a significantly higher average general mental health than men, which is interpreted as women having worse general mental health compared to men (17.77 ± 5.85 v/s 15.75 ± 6.67; t (380) = 9.242, p = 0.003, r2 = 0.42, with a small effect size).”

Similarly, for the legend of Table 1 and Figure 1, page 7 and 9 respectively. now reads as follows:

“The general mental health variable on the GHQ-12 scale, the higher the score, the worse general mental health.”

Comment 18. Table 2: Remove PANAS as variable label. PANAS usually refers to the Positive and Negative Affect Schedule by Watson et al. (1988).

Response 18: We agree with the reviewer's comment that the abbreviation PANAS creates confusion. The abbreviation PANAS in Table 2 was replaced by positive and negative affective experience. In this way, it is also consistent with what is mentioned in section 2.2 Procedure and instruments, and with the rest of the report in section 3. Results of the manuscript.

Comment 19: Lines 281 – 292: Statement of findings goes beyond what the statistical analyses support. Revise and be clear as to what the results truly support versus what is speculation or conjecture, and thus needs further testing in future studies.

Response 19: We appreciate the reviewer's comment that allowed us to improve the wording of the first paragraph of the Discussion of the manuscript. In this area, we agree with the reviewer that the way we have worded this paragraph could lead to confusion, in such a way that our assertions and conjectures seem to be beyond the support of statistical analysis. Therefore, the first paragraph of the discussion is modified, emphasizing that the findings correspond to Chilean university students who participated in this study.

“The main findings of the behavioral profiles established for the Chilean university students who participated in this study showed that those who presented a physically in-active and sedentary behavioral profile during the COVID-19 pandemic experienced worse well-being, positive affective experiences and general mental health. When adjusting for confounding variables, students who presented a physically active and non-sedentary profile were associated with better general mental health than those who are physically active and sedentary. Additionally, in relation to mental health symptoms, the behavior profile was related to fatigue, changes in eating behavior, and concentration problems, presenting a higher prevalence in students with sedentary behavior profiles regardless of behavior related to physical activity. On the contrary, those who had a physically active behavior profile presented a lower prevalence of mental health symptoms regardless of their sedentary time.”

Comment 20. Lines 323 – 326: Unclear statement. Make sure not to extrapolate beyond the results of the analyses unless it is clear conjecture and preparation for future exploration.

Response 20: We have removed the statement "an unexamined issue in other studies in the area". In this way, we believe that the statement belonging to the discussion part, indicated by the reviewer, can be read more clearly. Now, the modified statement that belongs to the fifth paragraph of the discussion (page 12-13) reads like this

“Therefore, the scientific strength of the present publication lies in its design, by contributing new knowledge to the literature being the first study that considers the comparison between physical activity and sedentarism in young university students in times of COVID-19.”

Comment 21. Lines 361-362: Please include a reference for the use of measures of facial and non-verbal expression for well-being.

Response 21: To respond to reviewer comment 21, we have incorporated citation number [43], which we have previously used in the paragraph to which this sentence belongs, as well as in the introduction to this manuscript. In this way, the statement indicated by the reviewer was modified to read as follows:

“For example, measures of facial and non-verbal expression in the case of well-being [43], and for physical activity the use of accelerometers, which would allow a more precise recording of planned and unplanned physical activity [81].”

  1. Diener E, Pressman SD, Hunter J, Delgadillo-Chase D. If, Why, and When Subjective Well-Being Influences Health, and Future Needed Research. Appl Psychol Heal Well-Being [Internet]. 2017 Jul 1 [cited 2021 Jun 26];9(2):133–67. Available from: 10.1111/aphw.12090

Comment 22. Lines 372 - 376: Sentence is unclear. There may be punctuation missing.

Response 22: Thanks to the reviewer's comment, we have improved the wording in the Discussion section of our manuscript. Consequently, the sentence belonging to the last paragraph of the discussion (page 13-14), which was modified, now reads as follows:

“In addition, this study allowed generating a pattern of subjective well-being and mental health according to the different behaviors of physical activity and sedentary lifestyle of university students, which could help understand which groups of students have the worst indicators of well-being and mental health”

Round 2

Reviewer 1 Report

The revision has addressed most of the points that I raised in my previous review. Until the final publication of the manuscript, I would encourage the authors to deepen the discussion in Section 5 by highlighting your findings and contribution, especially in reference to the previous literature.

Author Response

Comment. The revision has addressed most of the points that I raised in my previous review. Until the final publication of the manuscript, I would encourage the authors to deepen the discussion in Section 5 by highlighting your findings and contribution, especially in reference to the previous literature.

Response: We fully agree with the reviewer's comment that we needed to strengthen the discussion section. Therefore, we followed the suggestion given by the reviewer that it was necessary to highlight the findings and contributions of the study, especially in reference to previous literature. This led us to make important modifications to the discussion. In particular, we have added sentences about the findings that support hypothesis 1 of our study, and the contrast of these results with the supporting evidence. We have also discussed the findings regarding hypothesis 2 of the study. The protective effect of physical activity on well-being and mental health during the pandemic in sedentary students. And finally, as a sedentary lifestyle seems to be one of the variables that most affects the mental health of university students in Chile during the pandemic.

In addition, because of modifications in the discussion, as well as the reviewer's suggestion to highlight our findings, we were led to modify the conclusion of the abstract section.

On the other hand, information was added to strengthen the contributions section of the study.

Therefore, the first four paragraphs of the discussion section have been changed to read as follows:

“This study aimed to analyze the relationship between the behavioral profile of physical activity and sedentary time with subjective well-being and mental health in university students during the COVID-19 pandemic in Chile. The main findings indicate that Chilean university students who presented a physically inactive and sedentary profile during the COVID-19 pandemic experienced worse well-being, positive affective experiences, and general mental health. When adjusting for confounding variables, students who presented a physically active and non-sedentary profile were associated with better general mental health than those who are physically active and sedentary. Additionally, in relation to mental health symptoms, the behavior profile was related to fatigue, changes in eating behavior, and concentration problems, presenting a higher prevalence in students with sedentary behavior profiles regardless of behavior related to physical activity. On the contrary, those who had a physically active behavior profile presented a lower prevalence of mental health symptoms regardless of their sedentary time.

These findings support hypothesis 1 raised in this study, that physically inactive and sedentary college students have worse levels of subjective well-being and mental health compared to their physically active and non-sedentary peers. Similarly, these results are consistent with studies on the behavior of physical activity and sedentary time and its association with well-being and mental health in university students during the COVID-19 pandemic [40,45,52,72]. In this area, a study by Pengpid et al. in 12,492 university students from 24 countries, reported that physically active and non-sedentary students presented better life satisfaction, happiness and perceived health status during the COVID -19 pandemic [40]. Likewise, a study of 255 university students in the United Kingdom during the COVID-19 pandemic showed how well-being and physical activity decreased, while stress and sedentary lifestyle increased [72].

Regarding hypothesis 2 raised in this study, the results do not provide sufficient evidence to support that physically active and sedentary student presented lower subjective well-being and mental health than their physically active and non-sedentary peers. However, when analyzing mental health symptoms, differences were observed between these behavioral profiles. Thus, physically active and non-sedentary students had a lower prevalence of mental health symptoms such as fatigue, changes in eating behavior, and concentration problems. The foregoing, in contrast to what was presented by active but sedentary students, who presented a higher prevalence of these mental health symptoms. In this sense, similar results were observed by Rees-Punia et al. in which participants who increased their sedentary lifestyle, became inactive, or decreased their moderate to vigorous activity were more likely to experience depression related to psychological distress during the COVID-19 pandemic  [73].

In addition, a particular finding could be observed regarding the mental health outcomes of both sedentary and non-sedentary physically active students, who reported significantly better general mental health than inactive and sedentary students. Specifically, a protective effect of physical activity on the mental health of physically active but sedentary students was observed. However, this protective effect disappears when considering the confounding variables, gender, symptoms of COVID-19, sports time band, access to green areas, and study program. Whereas only physically active and non-sedentary students continue to maintain significantly better general mental health than inactive and sedentary students. In this sense, the study by Haider et al. found that the presence of high levels of physical activity was associated with greater well-being and fewer symptoms of depression and anxiety in times of the COVID-19 pandemic [74]. Furthermore, Fornili et al. reported that, during confinement, severe levels of anxiety or depression were found in 240,000 students, with a high probability of being severely anxious or depressed for those who stopped practicing their usual practice of physical activity [75].”

 Similarly, the abstract conclusion was modified and now reads:

“Chilean university students with a physically inactive and sedentary profile during the pandemic presented worse well-being and mental health, with a sedentary lifestyle being one of the variables that most affects the mental health of these students. Therefore, measures should be implemented to encourage this population to maintain adequate levels of physical activity and reduce sedentary times.”

Finally, the practical implication was modified and now reads:

On the other hand, one of the practical implications of the present study for society lies in providing updated evidence on the detrimental effects of physical inactivity and sedentary behavior on the emotional state of university students. These findings can serve as a guide for university authorities to promote programs to detect risk factors in the mental health of students, as well as educational programs aimed at not minimizing the adverse effects of sedentary lifestyle. and its consequences on university students’ various emotional states.

Reviewer 2 Report

Authors adequately addressed the primary concerns of the manuscript. Thank you for your thorough edits. 

Author Response

Thank you very much for all your comments, they helped us to improve the manuscript.